# STABILITY AND INFORMATION RETRIEVAL:
# PRECISE ANALYSIS FOR RANDOM AND NTK FEATURES

## ABSTRACT

Deep learning models are able to memorize the training set. This makes them vulnerable to recovery attacks, raising privacy concerns to users, and many widespread algorithms such as empirical risk minimization (ERM) do not directly enforce safety guarantees. In this paper, we study the safety of ERM models when the training samples are interpolated (i.e., *at interpolation*) against a family of powerful black-box information retrieval attacks. Our analysis quantifies this safety via two separate terms: *(i)* the model *stability* with respect to individual training samples, and *(ii)* the *feature alignment* between attacker query and original data. While the first term is well established in learning theory and it is connected to the generalization error in classical work, the second one is, to the best of our knowledge, novel. Our key technical result characterizes precisely the feature alignment for the two prototypical settings of random features (RF) and neural tangent kernel (NTK) regression. This indicates that the attack weakens with an increase in generalization capability, unveiling the role of the model and of its activation function. Numerical experiments show an agreement with our theory not only for RF/NTK models, but also for deep neural networks trained on standard datasets (MNIST, CIFAR-10).

## 1   INTRODUCTION

Deep learning models can memorize the training dataset (Shokri et al., 2017), which becomes concerning if sensitive information can be extracted by adversarial users. Thus, a thriving research effort has aimed at addressing this issue, with differential privacy (Dwork & Roth, 2014) emerging as a safety criterion. The workhorse of this framework is the differentially private stochastic gradient descent (DPSGD) algorithm (Abadi et al., 2016), which exploits a perturbation of the gradients to reduce the influence of a single sample on the final trained model. Despite numerous improvements and provable privacy guarantees (Andrew et al., 2021), this approach still comes at a significant performance cost (Tramer & Boneh, 2021), creating a difficult trade-off for users and developers. For this reason, many popular applications still rely on *empirical risk minimization* (ERM), with training times long enough to achieve 0 training error. These settings, however, do not offer any theoretical guarantee for privacy protection and, in particular, towards a family of powerful black-box attacks in which an external user aims to recover information about an individual training sample, without access to the model weights. This leads to the following critical questions:

*When are ERM-trained models resistant to information retrieval attacks?*
*How does this resistance depend on the model design and on its generalization performance?*

In this work, we focus on settings in which the attacker has partial knowledge about a training sample $z_1$ and aims to recover information about the rest; this is of particular interest when the training samples contain both public and private information, and it is considered in Bombari et al. (2022a), under the name of *relational privacy*. For concreteness, one can think of $z_1$ as an image with a specific individual in the foreground and a compromising background, or the combination of the name of a person with the relative home address. Designing the attack might be as easy as querying the right prompt in the language model fine-tuned over sensitive data (as empirically shown in Bombari et al. (2022a) on question answering tasks), and it does not require additional information, such as the rest of the dataset (Nasr et al., 2018) or the model weights (Nasr et al., 2019).

Formally, the samples are modeled by two distinct components, *i.e.*, $z \equiv [x, y]$. Given knowledge on $y$, the attacker aims to retrieve information about $x$ by querying the trained model with the *masked* sample $z^m := [-, y]$, see Figure 2 for an illustration. We consider generalized linear models trained with ERM, when the training algorithm completely fits the dataset. It turns out that in this setting the power of the attack can be *exactly* analyzed through two distinct components:

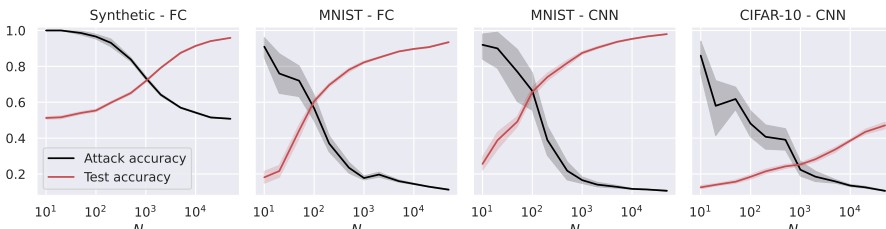

Figure 1: Test and attack accuracies as a function of the number of training samples $N$, for fully connected (FC, first two plots) and small convolutional neural networks (CNN, last two plots). In the first plot, we use synthetic (Gaussian) data with $d = 1000$, and the labeling function is $g(x) = \text{sign}(u^\top x)$. As we consider binary classification, the accuracy of random guessing is $0.5$. The other plots use subsets of the MNIST and CIFAR-10 datasets, with an external layer of noise added to images, see Figure 2. As we consider 10 classes, the accuracy of random guessing is $0.1$. We plot the average over 10 independent trials and the confidence band at 1 standard deviation.

1. The *feature alignment* $\mathcal{F}(z^m, z)$, see (7). This captures the similarity in feature space between the training sample $z$ and its masked counterpart $z^m$, and it depends on the feature map of the model. To the best of our knowledge, this is the first time that attention is raised over such an object.

2. The *stability* $\mathcal{S}_z$ of the model with respect to $z$, see Definition 3.1. Similar notions of stability are in the seminal work by Bousquet & Elisseeff (2002), which draws a connection to generalization.

Our technical contributions can be summarized as follows:

- We connect the stability of generalized linear models to the feature alignment between samples, see Lemma 4.1. Then, we show that this connection makes the attack resistance of the model a natural consequence of its generalization capability, when $\mathcal{F}(z^m, z)$ can be well approximated by a constant $\gamma > 0$, independent of the original sample $z$.

- We focus on two settings widely analyzed in the theoretical literature, *i.e.*, *(i)* random features (RF) (Rahimi & Recht, 2007), and *(ii)* the neural tangent kernel (NTK) (Jacot et al., 2018). Here, under a natural scaling of the models, we prove the concentration of $\mathcal{F}(z^m, z)$ to a positive constant $\gamma$, see Theorems 5.4 and 6.3. For the NTK, we obtain a closed-form expression for $\gamma$, which connects the power of the attack to the activation function.

We experimentally show that both synthetic and standard datasets (MNIST, CIFAR-10) agree well with the theoretical predictions. Remarkably, Figure 1 demonstrates the proportionality between test error and attack accuracy for various neural network architectures. This provides empirical evidence of the wide generality of our findings, which appears to go beyond RF/NTK models.

In a nutshell, our results give a precise characterization of the feature alignment of RF and NTK models. This in turn unveils how the accuracy of the attack grows with the generalization error, capturing the role of the model (and, specifically, of its activation). In contrast with the vast body of work relating differential privacy with generalization (Dwork et al., 2015b;a; Bassily et al., 2021; Steinke & Zakynthinou, 2020), we focus on the widespread paradigm of empirical risk minimization, when all the training samples are interpolated. In this setting, there is no explicit assumption on algorithmic stability and no a-priori guarantee in terms of privacy, as training is not performed via a differentially private mechanism. Thus, no generalization bound similar to Bousquet & Elisseeff (2002); Dwork et al. (2015b) can be explicitly computed. Furthermore, as we consider the interpolation regime, arguments based on the uniform stability of stochastic gradient methods (Hardt et al., 2016; Bassily et al., 2020) cannot be applied as well.

## 2 RELATED WORK

**Private machine learning.** Information retrieval via partial knowledge of the data is observed in question answering tasks by Bombari et al. (2022a). This setting is natural in language models, as they are prone to memorize the training set (Carlini et al., 2021; 2019), and to hallucinate it at test time (Zhao et al., 2020; Raunak et al., 2021). Differential privacy (Dwork & Roth, 2014) enables training deep learning models maintaining privacy guarantees. This is achieved through the DPSGD algorithm (Abadi et al., 2016) which, despite improvements (Yu et al., 2021; Andrew et al., 2021), still comes at a steep performance cost (Abadi et al., 2016; Tramer & Boneh, 2021). To circumvent the problem, a recent line of work (Boedihardjo et al., 2022a;b; 2023; He et al., 2023) utilizes synthetic datasets, analyzing efficient algorithms via tools from high dimensional probability.

**Stability.**  In more generality, it is linked to generalization in (Bassily et al., 2021; Elisseeff & Pontil, 2002; Mukherjee et al., 2006), and a wide range of variations on this object is discussed by Bousquet & Elisseeff (2002). Feldman (2020) takes a probabilistic viewpoint and shows that, when the data distribution is heavy-tailed, stability might be detrimental for learning. This is also supported empirically by Feldman & Zhang (2020). In contrast, Mukherjee et al. (2006) proves that, if the generalization gap vanishes with the number of samples, the learning algorithm has to be leave-one-out stable.

**Random Features and Neural Tangent Kernel.** Random features (RF) are introduced by Rahimi & Recht (2007) and can be regarded as a two-layer neural network with random first layer weights. This model is analytically tractable and offers deep-learning-like behaviours, such as double descent (Mei & Montanari, 2022). The neural tangent kernel (NTK) can be regarded as the kernel obtained by linearizing a neural network around the initialization (Jacot et al., 2018). A popular line of work has analyzed its spectrum (Fan & Wang, 2020; Adlam & Pennington, 2020; Wang & Zhu, 2021) and bounded its smallest eigenvalue (Soltanolkotabi et al., 2018; Nguyen et al., 2021; Montanari & Zhong, 2022; Bombari et al., 2022b). The behavior of the NTK is closely related to the memorization (Montanari & Zhong, 2022), optimization (Allen-Zhu et al., 2019; Du et al., 2019), generalization (Arora et al., 2019) and adversarial robustness (Bombari et al., 2023) of deep neural networks.

## 3  PRELIMINARIES

**Notation.** Given a vector $v$, we denote by $\|v\|_2$ its Euclidean norm. Given $v \in \mathbb{R}^{d_v}$ and $u \in \mathbb{R}^{d_u}$, we denote by $v \otimes u \in \mathbb{R}^{d_v d_u}$ their Kronecker product. Given a matrix $A \in \mathbb{R}^{m \times n}$, we denote by $P_A \in \mathbb{R}^{n \times n}$ the projector over $\mathrm{Span}\{\mathrm{rows}(A)\}$. All the complexity notations $\Omega(\cdot)$, $\mathcal{O}(\cdot)$, $o(\cdot)$ and $\Theta(\cdot)$ are understood for sufficiently large data size $N$, input dimension $d$, number of neurons $k$, and number of parameters $p$. We indicate with $C, c > 0$ numerical constants, independent of $N, d, k, p$.

**Setting.** Let $(Z, G)$ be a labelled training dataset, where $Z = [z_1, \ldots, z_N]^\top \in \mathbb{R}^{N \times d}$ contains the training data (sampled i.i.d. from a distribution $\mathcal{P}_Z$) on its rows and $G = (g_1, \ldots, g_N) \in \mathbb{R}^N$ contains the corresponding labels. We assume the label $g_i$ to be a deterministic function of the sample $z_i$. Let $\varphi : \mathbb{R}^d \to \mathbb{R}^p$ be a generic feature map, from the input space to a feature space of dimension $p$. We consider the following *generalized linear model*

$$f(z, \theta) = \varphi(z)^\top \theta, \tag{1}$$

where $\varphi(z) \in \mathbb{R}^p$ is the feature vector associated with the input sample $z$, and $\theta \in \mathbb{R}^p$ are the trainable parameters of the model. We minimize the empirical risk with a quadratic loss:

$$\min_\theta \left\| \varphi(Z)^\top \theta - G \right\|_2^2. \tag{2}$$

Here, $\varphi(Z) \in \mathbb{R}^{N \times p}$ is the feature matrix, containing $\varphi(z_i)$ in its $i$-th row. We use the shorthands $\Phi := \varphi(Z)$ and $K := \Phi \Phi^\top \in \mathbb{R}^{N \times N}$, where $K$ denotes the kernel associated with the feature map. If $K$ is invertible (i.e., the model can fit any set of labels $G$), gradient descent converges to the interpolator which is the closest in $\ell_2$ norm to the initialization (Gunasekar et al., 2017), i.e.,

$$\theta^* = \theta_0 + \Phi^+ (G - f(Z, \theta_0)), \tag{3}$$

where $\theta^*$ is the gradient descent solution, $\theta_0$ is the initialization, $f(Z, \theta_0) = \Phi^\top \theta_0$ the output of the model (1) at initialization, and $\Phi^+ := \Phi^\top K^{-1}$ the Moore-Penrose inverse. Let $z \sim \mathcal{P}_Z$ be an independent test sample. Then, we define the *generalization error* of the trained model as

$$\mathcal{R} = \mathbb{E}_{z \sim \mathcal{P}_Z} \left[ (f(z, \theta^*) - g)^2 \right], \tag{4}$$

where $g$ denotes the ground-truth label of the test sample $z$.

**Stability.** For our discussion, it is convenient to introduce quantities related to "incomplete" datasets. In particular, we indicate with $\Phi_{-1} \in \mathbb{R}^{(N-1) \times p}$ the feature matrix of the training set *without* the first sample $z_1$. For simplicity, we focus on the removal of the first sample, and similar considerations hold for the removal of any other sample. In other words, $\Phi_{-1}$ is equivalent to $\Phi$, without the first row. Similarly, using (3), we indicate with $\theta^*_{-1} := \theta_0 + \Phi^+_{-1} (G_{-1} - f(Z_{-1}, \theta_0))$ the set of parameters the algorithm would have converged to if trained over $(Z_{-1}, G_{-1})$, the original dataset without the first pair sample-label $(z_1, g_1)$. We can now proceed with the definition of our notion of "stability".

**Definition 3.1.** *Let $\theta^*$ ($\theta^*_{-1}$) be the parameters of the model $f$ given by* (1) *trained on the dataset $Z$ ($Z_{-1}$), as in* (3). *We define the* stability $\mathcal{S}_{z_1} : \mathbb{R}^d \to \mathbb{R}$ *with respect to the training sample $z_1$ as*

$$\mathcal{S}_{z_1} := f(\cdot, \theta^*) - f(\cdot, \theta^*_{-1}). \tag{5}$$

This quantity indicates how the trained model changes if we add $z_1$ to the dataset $Z_{-1}$. If the training algorithm completely fits the data (as in (3)), then $\mathcal{S}_{z_1}(z_1) = g_1 - f(z_1, \theta^*_{-1})$, which implies that

$$\mathbb{E}_{z_1 \sim \mathcal{P}_Z}\left[\mathcal{S}^2_{z_1}(z_1)\right] = \mathbb{E}_{z_1 \sim \mathcal{P}_Z}\left[\left(f(z_1, \theta^*_{-1}) - g_1\right)^2\right] = \mathbb{E}_{z \sim \mathcal{P}_Z}\left[\left(f(z, \theta^*_{-1}) - g\right)^2\right] =: \mathcal{R}_{Z_{-1}}, \tag{6}$$

where the purpose of the second step is just to match the notation used in (4), and $\mathcal{R}_{Z_{-1}}$ denotes the generalization error of the algorithm that uses $Z_{-1}$ as training set.

Bousquet & Elisseeff (2002) show that different notions of stability imply upper bounds on the generalization gap. In particular, if the cost function is limited from above, Theorem 11 in (Bousquet & Elisseeff, 2002) proves that the condition $\mathbb{E}_Z\left[\mathcal{S}^2_{z_1}(z_1)\right] \leq \beta$ ($\beta$-*pointwise hypothesis stability*, see Definition 4) gives an upper bound on the generalization error. Our viewpoint is different, since we do not study the generalization error of stable algorithms, as done *e.g.* in (Bassily et al., 2021; Elisseeff & Pontil, 2002; Mukherjee et al., 2006). In contrast, our goal is to characterize the power of information retrieval attacks, without any stability assumption on the training algorithm.

**Reconstruction attack.** Let the input samples be decomposed in two *independent* components, *i.e.*, $z \equiv [x, y]$. With this notation, we mean that $z \in \mathbb{R}^d$ is the concatenation of $x \in \mathbb{R}^{d_x}$ and $y \in \mathbb{R}^{d_y}$ ($d_x + d_y = d$). Here, $x$ is the part of the input that is useful to accomplish the task (*e.g.*, the cat in top-left image of Figure 2), while $y$ is noise (*e.g.*, the background). Formally, we assume that, for $i \in \{1, \dots, N\}$, $g_i = g(x_i)$, where $g$ is a deterministic labelling function, *i.e.*, the label depends only on $x$ and it is independent of $y$. In practice, the algorithm may overfit to the noise component, learning the spurious correlations between $y_i$ and the corresponding label $g_i$. An attacker might then exploit this phenomenon to reconstruct the label $g_i$, by simply querying the model with the noise component $y_i$. Without access to the model, this reconstruction would be impossible, as the noise $y_i$ is independent from $x_i$, and therefore from $g(x_i)$. In our theoretical analysis, we assume the attacker to have access to a *masked* sample $z^m_i(x) = [x, y_i]$, *i.e.*, a version of $z_i$ in which the component $x_i$ is replaced with an independent sample $x$ taken from the same distribution. We do the same in the synthetic setting of the experiments, while for MNIST and CIFAR-10 we just set $x$ to 0, see Figure 2. Our goal is to understand whether the output of the model evaluated on such query, *i.e.*, $f(z^m_i, \theta^*)$, provides information on the ground-truth label $g(x_i)$. As the setting is symmetric with respect to the data ordering, without loss of generality, we assume the attack to be aimed towards the first sample $z_1$.

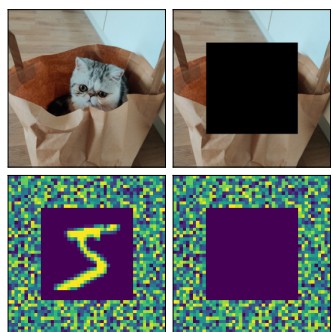

Figure 2: Example of a training sample $z$ (top-left) and its masked counterpart $z^m$ (top-right). In experiments, we add a noise background ($y$) around the original images ($x$) before training (bottom-left). The attack consists in querying the trained model only with the noise component (bottom-right).

The setting above in which an attacker tries to recover information on $x_i$ from $y_i$ is a known issue (Bombari et al., 2022a), and it is common when the sensitive information is not in the data itself, but rather in the relation among data points. A first motivating example comes from face recognition in computer vision. If the attacker wants to know if a certain individual ($x$) was at a certain compromising location ($y$), they could simply plug in the trained model a picture of the location without the individual (the empty shopping bag, in the first example of Figure 2). A second motivating example comes from NLP. Sensitive information ($x$) about an individual ($y$) is stored in a textual dataset. The attacker could guess that $y$ was mentioned in the training dataset, and can try to recover $x$ via the prompt *"The address of $y$ is..."*. Similar experiments are performed for question-answering in Bombari et al. (2022a), where the tokens containing the information which is relevant to solve the task are masked, but the trained model can still hallucinate the correct answer.

## 4 STABILITY, GENERALIZATION AND INFORMATION RETRIEVAL

**Stability and feature alignment.** Our goal is to quantify how much information about $g(x_1)$ the attacker can recover through a generic query $z$. To do so, we relate $f(z, \theta^*)$ to the model evaluated

on the original sample $z_1$. It turns out that, for generalized linear regression, under mild conditions on the feature map $\varphi$, this can be elegantly done via the notion of *stability* of Definition 3.1.

**Lemma 4.1.** *Let $\varphi : \mathbb{R}^d \to \mathbb{R}^p$ be a generic feature map, such that the induced kernel $K \in \mathbb{R}^{N \times N}$ on the training set is invertible. Let $z_1 \in \mathbb{R}^d$ be an element of the training dataset $Z$, and $z \in \mathbb{R}^d$ a generic test sample. Let $P_{\Phi_{-1}}$ be the projector over $Span\{\mathrm{rows}(\Phi_{-1})\}$, and $\mathcal{S}_{z_1}$ be the stability with respect to $z_1$, as in Definition 3.1. Let us denote by*

$$\mathcal{F}_\varphi(z, z_1) := \frac{\varphi(z)^\top P_{\Phi_{-1}}^\perp \varphi(z_1)}{\left\| P_{\Phi_{-1}}^\perp \varphi(z_1) \right\|_2^2} \tag{7}$$

*the* feature alignment *between $z$ and $z_1$. Then, we have*

$$\mathcal{S}_{z_1}(z) = \mathcal{F}_\varphi(z, z_1)\, \mathcal{S}_{z_1}(z_1). \tag{8}$$

The idea of the argument is to express $P_\Phi$ as $P_{\Phi_{-1}}$ plus the projector over the span of $P_{\Phi_{-1}}^\perp \varphi(z_1)$, by leveraging the Gram-Schmidt decomposition of $P_\Phi$. Lemma 4.1 highlights the role of the projector $P_{\Phi_{-1}}$ to study the stability of over-parameterized generalized linear models with respect to $z_1$. Classical work (Hoaglin & Welsch, 1978) tackles the same problem in the under-parametrized regime, exploiting the projector $H = \Phi(\Phi^\top \Phi)^{-1}\Phi^\top \in \mathbb{R}^{N \times N}$ ($p \leq N$ is needed for $\Phi^\top \Phi$ to be invertible), known in the literature as the *hat matrix*. The qualitatively different behaviour of the two regimes requires the proof of Lemma 4.1 to follow a different strategy, which is discussed in Appendix B.

In words, Lemma 4.1 relates the stability with respect to $z_1$ evaluated on the two samples $z$ and $z_1$ through the quantity $\mathcal{F}_\varphi(z, z_1)$, which captures the similarity between $z$ and $z_1$ in the feature space induced by $\varphi$. As a sanity check, the feature alignment between any sample and itself is equal to one, which trivializes (8). Then, as $z$ and $z_1$ become less aligned, the stability $\mathcal{S}_{z_1}(z) = f(z, \theta^*) - f(z, \theta_{-1}^*)$ starts to differ from $\mathcal{S}_{z_1}(z_1) = f(z_1, \theta^*) - f(z_1, \theta_{-1}^*)$, as described quantitatively by (8). We note that the feature alignment also depends on the rest of the training set $Z_{-1}$, as $Z_{-1}$ implicitly appears in the projector $P_{\Phi_{-1}}^\perp$. We also remark that the invertibility of $K$ directly implies that the denominator in (7) is different from zero, see Lemma B.1.

**Generalization and power of the attack.** Armed with Lemma 4.1, we now characterize the power of the attack query $f(z_1^m, \theta^*)$. Let us replace $\mathcal{F}_\varphi(z_1^m, z_1)$ in (8) with a constant $\gamma_\varphi > 0$, independent from $z_1$. This is justified by Sections 5 and 6, where we prove the concentration of $\mathcal{F}_\varphi(z_1^m, z_1)$ for the RF and NTK model, respectively. Then, by using the definition of stability in (5), we get

$$f(z_1^m, \theta^*) = f(z_1^m, \theta_{-1}^*) + \gamma_\varphi\, \mathcal{S}_{z_1}(z_1) = f(z_1^m, \theta_{-1}^*) + \gamma_\varphi\left(g_1 - f(z_1, \theta_{-1}^*)\right). \tag{9}$$

Note that $f(z_1^m, \theta_{-1}^*)$ is independent from $g_1$, as it doesn't depend on $x_1$. Thus, if the algorithm is stable, in the sense that $\mathcal{S}_{z_1}(z_1)$ is small, then it is also private, as $f(z_1^m, \theta^*)$ has little dependence on $g_1$. Conversely, if $\mathcal{S}_{z_1}(z_1)$ grows, then $f(z_1^m, \theta^*)$ will start picking up the correlation with $g_1$. More concretely, we can look at the covariance between $f(z_1^m, \theta^*)$ and $g_1$, in the probability space of $z_1$:

$$\mathrm{Cov}\left(f(z_1^m, \theta^*), g_1\right) = \gamma_\varphi \mathrm{Cov}\left(\mathcal{S}_{z_1}(z_1), g_1\right) \leq \gamma_\varphi \sqrt{\mathrm{Var}\left(\mathcal{S}_{z_1}(z_1)\right)\mathrm{Var}\left(g_1\right)} \leq \gamma_\varphi \sqrt{\mathcal{R}_{Z_{-1}}}\sqrt{\mathrm{Var}\left(g_1\right)}. \tag{10}$$

Here, the first step uses (9) and the independence between $f(z_1^m, \theta_{-1}^*)$ and $g_1$, the second step is an application of Cauchy-Schwarz, and the last step follows from (6). Let us focus on the RHS of (10). While $\sqrt{\mathrm{Var}\left(g_1\right)}$ is a simple scaling factor, $\gamma_\varphi$ and $\sqrt{\mathcal{R}_{Z_{-1}}}$ lead to an interesting interpretation: we expect the attack to become more powerful as the similarity between $z_1^m$ and $z_1$ (formalized by $\mathcal{F}_\varphi(z_1^m, z_1)$) increases, and less effective as the generalization error of the model decreases. In fact, the potential threat hinges on the model overfitting the $y$-component at training time. This overfitting would both cause higher generalization error, and higher chances of recovering $g_1$ given only $y_1$.

The power of the attacker can be also characterized by the *recovery loss*, defined as

$$\mathcal{L} := \mathbb{E}_{z_1}\left[\left(f(z_1^m, \theta^*) - g_1\right)^2\right]. \tag{11}$$

By introducing the shorthands $\bar{f} := \mathbb{E}_{z_1}[f(z_1^m, \theta^*)]$ and $\bar{g} := \mathbb{E}_{z_1}[g_1]$, we have the lower bound:

$$\mathcal{L} = \mathbb{E}_{z_1}\left[\left(\left(f(z_1^m, \theta^*) - \bar{f}\right) - (g_1 - \bar{g}) + \left(\bar{f} - \bar{g}\right)\right)^2\right]$$

$$= \mathbb{E}_{z_1}\left[\left(f(z_1^m, \theta^*) - \bar{f}\right)^2\right] + \mathbb{E}_{z_1}\left[(g_1 - \bar{g})^2\right] + \mathbb{E}_{z_1}\left[\left(\bar{f} - \bar{g}\right)^2\right] - 2\,\mathrm{Cov}\left(f(z_1^m, \theta^*), g_1\right) \tag{12}$$

$$\geq \mathbb{E}_{z_1}\left[(g_1 - \bar{g})^2\right] - 2\,\gamma_\varphi \sqrt{\mathcal{R}_{Z_{-1}}}\sqrt{\mathrm{Var}\left(g_1\right)}.$$

The first term on the RHS of (12) is the minimal loss when the attacker does not have access to the model. In fact, in this case, the best choice to minimize $\mathcal{L}$ is to simply guess the expected value of $g_1$. The second term indicates how the attacker can improve the trivial guess $\bar{g}$. This term depends on $\mathcal{R}_{Z_{-1}}$ and $\gamma_\varphi$, according to (10). A lower bound on $\mathcal{L}$ that does not neglect the variance of $f(z_1^m, \theta^*)$ (i.e., the first term in the second line of (12)) can be obtained by assuming both the labels and the predictor to be balanced ($\mathbb{E}_{x_1}[g_1] = 0$ and $\mathbb{E}_x[f(z_1^m, \theta_{-1}^*)] = 0$):

$$\mathbb{E}_x[\mathcal{L}] \geq \mathbb{E}_x[\mathcal{L}_0] + \gamma_\varphi^2 \mathcal{R}_{Z_{-1}} - 2\gamma_\varphi \sqrt{\mathcal{R}_{Z_{-1}}} \sqrt{\mathrm{Var}(g_1)}, \tag{13}$$

where $\mathcal{L}_0 = \mathbb{E}_{x_1, y_1}[(f(z_1^m, \theta_{-1}^*) - g_1)^2]$ is the loss of an attacker with no information about the true label $g_1$. When the generalization error is small, the RHS of (13) is similar to that of (12). The proof of this result is deferred to Appendix F.

## 5 MAIN RESULT FOR RANDOM FEATURES

The *random features (RF) model* takes the form

$$f_{\mathrm{RF}}(z, \theta) = \varphi_{\mathrm{RF}}(z)^\top \theta, \qquad \varphi_{\mathrm{RF}}(z) = \phi(Vz), \tag{14}$$

where $V$ is a $k \times d$ matrix s.t. $V_{i,j} \sim_{\mathrm{i.i.d.}} \mathcal{N}(0, 1/d)$, and $\phi$ is an activation applied component-wise. The number of parameters of this model is $k$, as $V$ is fixed and $\theta \in \mathbb{R}^k$ contains trainable parameters.

**Assumption 5.1** (Data distribution). *The input data $(z_1, \ldots, z_N)$ are $N$ i.i.d. samples from $\mathcal{P}_Z = \mathcal{P}_X \times \mathcal{P}_Y$, such that $z_i \in \mathbb{R}^d$ can be written as $z_i = [x_i, y_i]$, with $x_i \in \mathbb{R}^{d_x}$, $y_i \in \mathbb{R}^{d_y}$ and $d = d_x + d_y$. We assume that $x_i \sim \mathcal{P}_X$ is independent of $y_i \sim \mathcal{P}_Y$, and the following holds:*

1. *$\|x\|_2 = \sqrt{d_x}$, and $\|y\|_2 = \sqrt{d_y}$, i.e., the data have normalized norm.*

2. *$\mathbb{E}[x] = 0$, and $\mathbb{E}[y] = 0$, i.e., the data are centered.*

3. *Both $\mathcal{P}_X$ and $\mathcal{P}_Y$ satisfy the Lipschitz concentration property.*

The first two assumptions can be achieved by pre-processing the raw data, and they could be relaxed as in Assumption 1 of Bombari et al. (2022b) at the cost of a more involved argument. The third assumption (see Appendix A for details) covers a number of important cases, e.g., standard Gaussian (Vershynin, 2018), uniform on the sphere/hypercube (Vershynin, 2018), or data obtained via GANs (Seddik et al., 2020). This requirement is common in the related literature (Bombari et al., 2022b; 2023; Bubeck & Sellke, 2021; Nguyen et al., 2021) and it is often replaced by a stronger requirement (e.g., data uniform on the sphere), see Montanari & Zhong (2022).

**Assumption 5.2** (Over-parameterization and high-dimensional data).

$$N \log^3 N = o(k), \qquad \sqrt{d} \log d = o(k), \qquad k \log^4 k = o(d^2). \tag{15}$$

The first condition in (15) requires the number of neurons $k$ to scale faster than the number of data points $N$. This over-parameterization leads to a lower bound on the smallest eigenvalue of the kernel induced by the feature map, which in turn implies that the model interpolates the data, as required to write (3). This over-parameterized regime also achieves minimum test error (Mei & Montanari, 2022). Combining the second and third conditions in (15), we have that $k$ can scale between $\sqrt{d}$ and $d^2$ (up to log factors). Finally, merging the first and third condition gives that $d^2$ scales faster than $N$. We notice that this holds for standard datasets (MNIST, CIFAR-10 and ImageNet).

**Assumption 5.3** (Activation function). *The activation function $\phi$ is a non-linear $L$-Lipschitz function.*

**Theorem 5.4.** *Let Assumptions 5.1, 5.2, and 5.3 hold, and let $x \sim \mathcal{P}_X$ be sampled independently from everything. Consider querying the trained RF model (14) with $z_1^m = [x, y_1]$. Let $\alpha = d_y/d$ and $\mathcal{F}_{\mathrm{RF}}(z_1^m, z_1)$ be the feature alignment between $z_1^m$ and $z_1$, as defined in (7). Then,*

$$|\mathcal{F}_{\mathrm{RF}}(z_1^m, z_1) - \gamma_{\mathrm{RF}}| = o(1), \tag{16}$$

*with probability at least $1 - \exp(-c \log^2 N)$ over $V$, $Z$ and $x$, where $c$ is an absolute constant, and $\gamma_{\mathrm{RF}} \leq 1$ does not depend on $z_1$ and $x$. Furthermore, we have*

$$\gamma_{\mathrm{RF}} > \frac{\sum_{l=2}^{+\infty} \mu_l^2 \alpha^l}{\sum_{l=1}^{+\infty} \mu_l^2} - o(1), \tag{17}$$

*with probability at least $1 - \exp(-c \log^2 N)$ over $V$, and $Z_{-1}$, where $c$ is an absolute constant,* i.e., *$\gamma_{\mathrm{RF}}$ is bounded away from $0$ with high probability.*

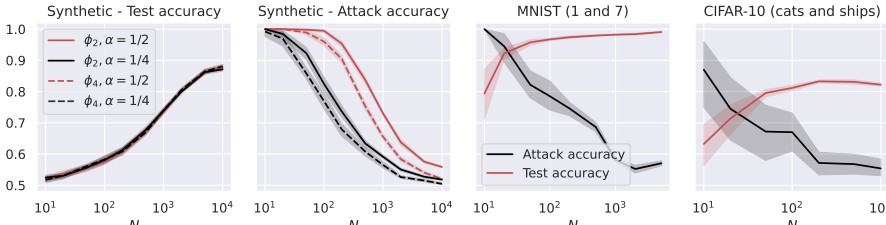

Figure 3: Test and attack accuracies as a function of the number of training samples $N$, for various binary classification tasks. In the first two plots, we consider the RF model in (14) with $k = 10^5$ trained over Gaussian data with $d = 1000$. The labeling function is $g(x) = \text{sign}(u^\top x)$. We repeat the experiments for $\alpha = \{0.25, 0.5\}$, and for the two activations $\phi_2 = h_1 + h_2$ and $\phi_4 = h_1 + h_4$, where $h_i$ denotes the $i$-th Hermite polynomial. In the last two plots, we consider the same model with ReLU activation, trained over two MNIST and CIFAR-10 classes. The width of the noise background is 10 pixels for MNIST and 8 pixels for CIFAR-10, see Figure 2. The reconstruction attack queries the model only with the noise background, replacing all the other pixels with 0, and takes the sign of the output. As we consider binary classification, an accuracy of 0.5 is achieved by random guessing. We plot the average over 10 independent trials and the confidence band at 1 standard deviation.

The combination of Theorem 5.4 and (10) shows that the more the algorithm is stable (and, therefore, capable of generalizing), the smaller is the statistical dependency between the query of the attacker and the true label. The proportionality constant $\gamma_{\text{RF}}$ is between $\frac{\sum_{l=2}^{+\infty} \mu_l^2 \alpha^l}{\sum_{l=1}^{+\infty} \mu_l^2} > 0$ and 1. In particular, the lower bound increases with $\alpha$ (as expected, since $\alpha$ represents the fraction of the input to which the attacker has access), and it depends in a non-trivial way on the activation function via its Hermite coefficients. These effects are clearly displayed in Figure 3 for binary classification tasks involving synthetic (first two plots) and standard (last two plots) datasets. Specifically, as the number of samples $N$ increases, the test accuracy increases and, correspondingly, the effect of the reconstruction attack decreases. Furthermore, for the synthetic dataset, while the test accuracy does not depend on $\alpha$ and on the activation function, the success of the attack increases with $\alpha$ and by taking an activation function with dominant low-order Hermite coefficients, as predicted by (17).

**Proof sketch.** We set $\gamma_{\text{RF}} := \mathbb{E}_{z_1, z_1^m}[\varphi_{\text{RF}}(z_1^m)^\top P_{\Phi_{-1}}^\perp \varphi_{\text{RF}}(z_1)] / \mathbb{E}_{z_1}[\|P_{\Phi_{-1}}^\perp \varphi_{\text{RF}}(z_1)\|_2^2]$. Here, $P_{\Phi_{-1}}$ denotes the projector over $\text{Span}\{\text{rows}(\Phi_{\text{RF}, -1})\}$ and $\Phi_{\text{RF}, -1}$ is the RF feature matrix after removing the first row. With this choice, the numerator and denominator of $\gamma_{\text{RF}}$ equal the expectations of the corresponding quantities appearing in $\mathcal{F}_{\text{RF}}(z_1^m, z_1)$. Thus, the concentration result in (16) is obtained from the general form of the Hanson-Wright inequality in (Adamczak, 2015), see Lemma D.7. The upper bound $\gamma_{\text{RF}} \leq 1$ follows from an application of Cauchy-Schwarz inequality. In contrast, the lower bound is more involved and it is obtained via the following three steps:

*Step 1: Centering the feature map $\varphi_{\text{RF}}$.* We extract the term $\mathbb{E}_V[\phi(Vz)]$ from the expression of $\mathcal{F}_{\text{RF}}$ and we show that it can be neglected, due to the specific structure of $P_{\Phi_{-1}}^\perp$. Specifically, letting $\tilde{\varphi}_{\text{RF}}(z) := \varphi_{\text{RF}}(z) - \mathbb{E}_V[\varphi_{\text{RF}}(z)]$, we have

$$\mathcal{F}_{\text{RF}}(z_1^m, z_1) \simeq \frac{\tilde{\varphi}_{\text{RF}}(z_1^m)^\top P_{\Phi_{-1}}^\perp \tilde{\varphi}_{\text{RF}}(z_1)}{\left\|P_{\Phi_{-1}}^\perp \tilde{\varphi}_{\text{RF}}(z_1)\right\|_2^2} = \frac{\tilde{\varphi}_{\text{RF}}(z_1^m)^\top \tilde{\varphi}_{\text{RF}}(z_1) - \tilde{\varphi}_{\text{RF}}(z_1^m)^\top P_{\Phi_{-1}} \tilde{\varphi}_{\text{RF}}(z_1)}{\left\|P_{\Phi_{-1}}^\perp \tilde{\varphi}_{\text{RF}}(z_1)\right\|_2^2}, \quad (18)$$

where $\simeq$ denotes an equality up to a $o(1)$ term. This is formalized in Lemma D.3.

*Step 2: Linearization of the centered feature map $\tilde{\varphi}_{\text{RF}}$.* We consider the terms $\tilde{\varphi}_{\text{RF}}(z_1^m), \tilde{\varphi}_{\text{RF}}(z_1)$ that multiply $P_{\Phi_{-1}}$ in the RHS of (18), and we show that they are well approximated by their first-order Hermite expansions ($\mu_1 V z_1^m$ and $\mu_1 V z_1$, respectively). In fact, the rest of the Hermite series scales at most as $N/d^2$, which is negligible due to Assumption 5.2. Specifically, Lemma D.4 implies

$$\frac{\tilde{\varphi}_{\text{RF}}(z_1^m)^\top \tilde{\varphi}_{\text{RF}}(z_1) - \tilde{\varphi}_{\text{RF}}(z_1^m)^\top P_{\Phi_{-1}} \tilde{\varphi}_{\text{RF}}(z_1)}{\left\|P_{\Phi_{-1}}^\perp \tilde{\varphi}_{\text{RF}}(z_1)\right\|_2^2} \simeq \frac{\tilde{\varphi}_{\text{RF}}(z_1^m)^\top \tilde{\varphi}_{\text{RF}}(z_1) - \mu_1^2 (V z_1^m)^\top P_{\Phi_{-1}}(V z_1)}{\left\|P_{\Phi_{-1}}^\perp \tilde{\varphi}_{\text{RF}}(z_1)\right\|_2^2}.$$

$$(19)$$

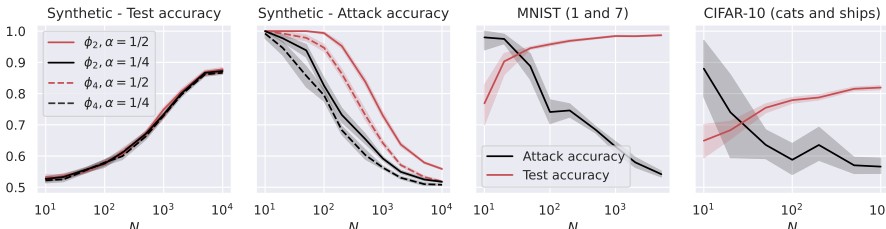

Figure 4: We consider the NTK model in (14) with $k = 100$ and, in the first two plots, we repeat the experiments for activations whose derivatives are $\phi_2' = h_0 + h_1$ and $\phi_4' = h_0 + h_3$, where $h_i$ denotes the $i$-th Hermite polynomial (see Appendix A.1). The rest of the setup is similar to that of Figure 3.

*Step 3: Lower bound in terms of $\alpha$ and $\{\mu_l\}_{l \geq 2}$.* To conclude, we express the RHS of (19) as follows:

$$
\begin{aligned}
&\frac{\tilde{\varphi}_{\text{RF}}(z_1^m)^\top \tilde{\varphi}_{\text{RF}}(z_1) - \mu_1^2(V z_1^m)^\top (V z_1) + \mu_1^2(V z_1^m)^\top P_{\Phi_{-1}}^\perp (V z_1)}{\|\tilde{\varphi}_{\text{RF}}(z_1)\|_2^2 - \left\|P_{\Phi_{-1}}\tilde{\varphi}_{\text{RF}}(z_1)\right\|_2^2} \\
&\gtrsim \frac{\tilde{\varphi}_{\text{RF}}(z_1^m)^\top \tilde{\varphi}_{\text{RF}}(z_1) - \mu_1^2(V z_1^m)^\top (V z_1)}{\|\tilde{\varphi}_{\text{RF}}(z_1)\|_2^2} \simeq \frac{\sum_{l=2}^{+\infty} \mu_l^2 \alpha^l}{\sum_{l=1}^{+\infty} \mu_l^2} > 0,
\end{aligned}
\tag{20}
$$

where $\gtrsim$ denotes an inequality up to a $o(1)$ term. As $P_{\Phi_{-1}} = I - P_{\Phi_{-1}}^\perp$, the term in the first line equals the RHS of (19). Next, we show that $\mu_1^2(V z_1^m)^\top P_{\Phi_{-1}}^\perp (V z_1)$ is equal to $\mu_1^2(V[y_1, 0])^\top P_{\Phi_{-1}}^\perp (V[y_1, 0])$ (which corresponds to the common noise part in $z_1, z_1^m$) plus a vanishing term, see Lemma D.5. As $\mu_1^2(V[y_1, 0])^\top P_{\Phi_{-1}}^\perp (V[y_1, 0]) \geq 0$, the inequality in the second line follows. The last step is obtained by showing concentration over $V$ of numerator and denominator. The expression on the RHS of (20) is strictly positive as $\alpha > 0$ and $\phi$ is non-linear by Assumption 5.3.

# 6 MAIN RESULT FOR NTK REGRESSION

We consider the following two-layer neural network

$$
f_{\text{NN}}(z, w) = \sum_{i=1}^{k} \phi\left(W_{i:} z\right).
\tag{21}
$$

Here, the hidden layer contains $k$ neurons; $\phi$ is an activation function applied component-wise; $W \in \mathbb{R}^{k \times d}$ denotes the weights of the hidden layer; $W_{i:}$ denotes the $i$-th row of $W$; and we set the $k$ weights of the second layer to 1. We indicate with $w$ the vector containing the parameters of this model, *i.e.*, $w = [\text{vec}(W)] \in \mathbb{R}^p$, with $p = kd$. We initialize the network with standard (*e.g.*, He's or LeCun's) initialization, *i.e.*, $[W_0]_{i,j} \sim_{\text{i.i.d.}} \mathcal{N}(0, 1/d)$. Now, the *NTK regression model* takes the form

$$
f_{\text{NTK}}(z, \theta) = \varphi_{\text{NTK}}(z)^\top \theta, \qquad \varphi_{\text{NTK}}(z) = \nabla_w f_{\text{NN}}(z, w)|_{w = w_0}.
\tag{22}
$$

Here, the vector of trainable parameters is $\theta \in \mathbb{R}^p$, with $p = kd$, which is initialized with $\theta_0 = w_0 = [\text{vec}(W_0)]$. This is the same model considered in Bombari et al. (2023); Dohmatob & Bietti (2022); Montanari & Zhong (2022), and $f_{\text{NTK}}(z, \theta)$ corresponds to the linearization of $f_{\text{NN}}(z, w)$ around the initial point $w_0$ (Bartlett et al., 2021; Jacot et al., 2018). An application of the chain rule gives

$$
\varphi_{\text{NTK}}(z) = z \otimes \phi'(W_0 z).
\tag{23}
$$

**Assumption 6.1** (Over-parameterization and topology).

$$
N \log^8 N = o(kd), \qquad N > d, \qquad k = \mathcal{O}(d).
\tag{24}
$$

The first condition is the smallest (up to $\log$ factors) over-parameterization that guarantees interpolation (Bombari et al., 2022b). The second condition is rather mild (it is easily satisfied by standard datasets) and purely technical. The third condition is required to lower bound the smallest eigenvalue of the kernel induced by the feature map, and a stronger requirement, *i.e.*, the strict inequality $k < d$, has appeared in prior work (Nguyen & Hein, 2017; 2018; Nguyen & Mondelli, 2020).

**Assumption 6.2** (Activation function). *The activation function $\phi$ is a non-linear function with $L$-Lipschitz first order derivative $\phi'$.*

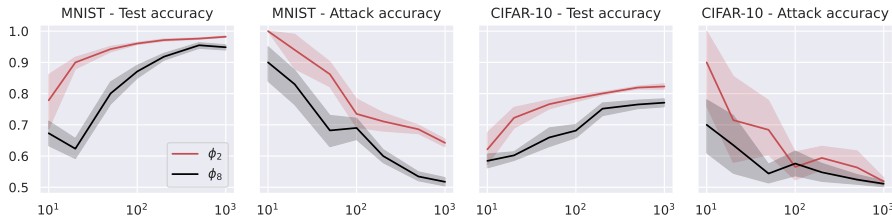

Figure 5: We consider the NTK model in (14) with $k = 100$, trained on MNIST (digits 1 and 7, first and second plots), and CIFAR-10 (cats and ships, third and fourth plots). We repeat the experiments for activations whose derivatives are $\phi_2' = h_0 + h_1$ and $\phi_8' = h_0 + h_7$, where $h_i$ denotes the $i$-th Hermite polynomial (see Appendix A.1). The rest of the setup is similar to that of Figure 3.

We denote by $\mu_l'$ the $l$-th Hermite coefficient of $\phi'$. We remark that the invertibility of the kernel $K_{\text{NTK}}$ induced by the feature map (22) follows from Lemma E.1. At this point, we are ready to state our main result for the NTK model, whose full proof is contained in Appendix E.

**Theorem 6.3.** *Let Assumptions 5.1, 6.1, and 6.2 hold, and let $x \sim \mathcal{P}_X$ be sampled independently from everything. Consider querying the trained NTK model* (22) *with $z_1^m = [x, y_1]$. Let $\alpha = d_y/d \in (0, 1)$ and $\mathcal{F}_{\text{NTK}}(z_1^m, z_1)$ be the feature alignment between $z_1^m$ and $z_1$, as defined in* (7). *Then,*

$$|\mathcal{F}_{\text{NTK}}(z_1^m, z_1) - \gamma_{\text{NTK}}| = o(1), \qquad where \quad 0 < \gamma_{\text{NTK}} := \alpha \frac{\sum_{l=1}^{+\infty} \mu_l'^2 \alpha^l}{\sum_{l=1}^{+\infty} \mu_l'^2} < 1, \qquad (25)$$

*with probability at least $1 - N \exp(-c \log^2 k) - \exp(-c \log^2 N)$ over $Z$, $x$, and $W$, where $c$ is an absolute constant.*

The combination of Theorem 6.3 and (10) connects stability with the statistical dependency between the attacker query and the true label, as done for the RF model. In addition, for the NTK model, we are able to express the limit $\gamma_{\text{NTK}}$ of the feature alignment in a closed form involving $\alpha$ and the Hermite coefficients of the derivative of the activation. The findings of Theorem 6.3 are clearly displayed in Figures 4 and 5: as $N$ increases, the test accuracy improves and the reconstruction attack becomes less effective; decreasing $\alpha$ or considering activations with dominant high-order Hermite coefficients also reduces the accuracy of the attack.

**Discussion.**     As (12) suggests, if the value of $\mathcal{F}_\varphi(z_1, z_1^m)$ converges to 0, no information can be recovered by the attacker. However, our Theorems 5.4 and 6.3 show that, for RF and NTK models respectively, a strictly positive geometric overlap ($\alpha > 0$) guarantees a strictly positive feature alignment. This in turn guarantees the attacker the possibility to recover information about the original data, as long as the generalization error is not 0, as discussed in (10). Our experiments both on synthetic and real datasets confirm these findings. First, in Figures 3, 4 and 5, we see a decrease in the attack accuracy as generalization improves, as discussed in Section 4. This behaviour is displayed also by neural networks, as shown in Figure 1. Additionally, our experiments confirm the characterizations of $\gamma_{\text{RF}}$ ($\gamma_{\text{NTK}}$) in terms of $\alpha$ and the Hermite coefficients of the activation (its derivative). The expected dependence is displayed also on real datasets (see Figure 5), where functions with higher order Hermite coefficients lead to models that are more resistant to the attack.

## 7   CONCLUSIONS

In this work, we study the accuracy of a family of powerful information recovery attacks for models trained via empirical risk minimization, in the interpolation regime. Our characterization hinges on *(i)* the classical notion of *stability* of the model w.r.t. a training sample, and *(ii)* a novel notion of *feature alignment* $\mathcal{F}(z_1, z_1^m)$ between the target of the attack $z_1$ and the adversarial query $z_1^m$. By providing a precise analysis for this feature alignment, we unveil a connection between generalization and privacy (intended in terms of the protection against such attacks). While the theoretical analysis focuses on generalized linear regression with random and NTK features, numerical results on different neural networks point to the generality of our findings, see Figure 1. We highlight that the formalism introduced by Lemma 4.1 applies to any feature map (e.g., with multiple fully-connected, convolutional or attention layers). Characterizing the feature alignment of such maps would allow to compare different models and establish which of those are the most resistant, paving the way to the principled design of privacy-preserving ML models.

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

## A  ADDITIONAL NOTATIONS AND REMARKS

Given a sub-exponential random variable $X$, let $\|X\|_{\psi_1} = \inf\{t > 0 \; : \; \mathbb{E}[\exp(|X|/t)] \leq 2\}$. Similarly, for a sub-Gaussian random variable, let $\|X\|_{\psi_2} = \inf\{t > 0 \; : \; \mathbb{E}[\exp(X^2/t^2)] \leq 2\}$. We use the analogous definitions for vectors. In particular, let $X \in \mathbb{R}^n$ be a random vector, then $\|X\|_{\psi_2} := \sup_{\|u\|_2=1} \|u^\top X\|_{\psi_2}$ and $\|X\|_{\psi_1} := \sup_{\|u\|_2=1} \|u^\top X\|_{\psi_1}$. Notice that if a vector has independent, mean 0, sub-Gaussian (sub-exponential) entries, then it is sub-Gaussian (sub-exponential). This is a direct consequence of Hoeffding's inequality and Bernstein's inequality (see Theorems 2.6.3 and 2.8.2 in Vershynin (2018)).

We say that a random variable or vector respects the Lipschitz concentration property if there exists an absolute constant $c > 0$ such that, for every Lipschitz continuous function $\tau : \mathbb{R}^d \to \mathbb{R}$, we have $\mathbb{E}|\tau(X)| < +\infty$, and for all $t > 0$,

$$\mathbb{P}\left(|\tau(x) - \mathbb{E}_X[\tau(x)]| > t\right) \leq 2e^{-ct^2/\|\tau\|_{\mathrm{Lip}}^2}. \tag{26}$$

When we state that a random variable or vector $X$ is sub-Gaussian (or sub-exponential), we implicitly mean $\|X\|_{\psi_2} = \mathcal{O}(1)$, i.e. it doesn't increase with the scalings of the problem. Notice that, if $X$ is Lipschitz concentrated, then $X - \mathbb{E}[X]$ is sub-Gaussian. If $X \in \mathbb{R}$ is sub-Gaussian and $\tau : \mathbb{R} \to \mathbb{R}$ is Lipschitz, we have that $\tau(X)$ is sub-Gaussian as well. Also, if a random variable is sub-Gaussian or sub-exponential, its $p$-th momentum is upper bounded by a constant (that might depend on $p$).

In general, we indicate with $C$ and $c$ absolute, strictly positive, numerical constants, that do not depend on the scalings of the problem, i.e. input dimension, number of neurons, or number of training samples. Their value may change from line to line.

Given a matrix $A$, we indicate with $A_{i:}$ its $i$-th row, and with $A_{:j}$ its $j$-th column. Given a square matrix $A$, we denote by $\lambda_{\min}(A)$ its smallest eigenvalue. Given a matrix $A$, we indicate with $\sigma_{\min}(A) = \sqrt{\lambda_{\min}(A^\top A)}$ its smallest singular value, with $\|A\|_{\mathrm{op}}$ its operator norm (and largest singular value), and with $\|A\|_F$ its Frobenius norm ($\|A\|_F^2 = \sum_{ij} A_{ij}^2$).

Given two matrices $A, B \in \mathbb{R}^{m \times n}$, we denote by $A \circ B$ their Hadamard product, and by $A * B = [(A_{1:} \otimes B_{1:}), \ldots, (A_{m:} \otimes B_{m:})]^\top \in \mathbb{R}^{m \times n^2}$ their row-wise Kronecker product (also known as Khatri-Rao product). We denote $A^{*2} = A * A$. We remark that $(A * B)(A * B)^\top = AA^\top \circ BB^\top$. We say that a matrix $A \in \mathbb{R}^{n \times n}$ is positive semi definite (p.s.d.) if it's symmetric and for every vector $v \in \mathbb{R}^n$ we have $v^\top A v \geq 0$.

### A.1  HERMITE POLYNOMIALS

In this subsection, we refresh standard notions on the Hermite polynomials. For a more comprehensive discussion, we refer to O'Donnell (2014). The (probabilist's) Hermite polynomials $\{h_j\}_{j \in \mathbb{N}}$ are an orthonormal basis for $L^2(\mathbb{R}, \gamma)$, where $\gamma$ denotes the standard Gaussian measure. The following result holds.

**Proposition A.1** (Proposition 11.31, O'Donnell (2014)). *Let $\rho_1, \rho_2$ be two standard Gaussian random variables, with correlation $\rho \in [-1, 1]$. Then,*

$$\mathbb{E}_{\rho_1, \rho_2}[h_i(\rho_1) h_j(\rho_2)] = \delta_{ij} \rho^i, \tag{27}$$

*where $\delta_{ij} = 1$ if $i = j$, and $0$ otherwise.*

The first 5 Hermite polynomials are

$$h_0(\rho) = 1, \quad h_1(\rho) = \rho, \quad h_2(\rho) = \frac{\rho^2 - 1}{\sqrt{2}}, \quad h_3(\rho) = \frac{\rho^3 - 3\rho}{\sqrt{6}}, \quad h_4(\rho) = \frac{\rho^4 - 6\rho^2 + 3}{\sqrt{24}}. \tag{28}$$

**Proposition A.2** (Definition 11.34, O'Donnell (2014)). *Every function $\phi \in L^2(\mathbb{R}, \gamma)$ is uniquely expressible as*

$$\phi(\rho) = \sum_{i \in \mathbb{N}} \mu_i^\phi h_i(\rho), \tag{29}$$

*where the real numbers $\mu_i^\phi$'s are called the Hermite coefficients of $\phi$, and the convergence is in $L^2(\mathbb{R}, \gamma)$. More specifically,*

$$\lim_{n \to +\infty} \left\| \left( \sum_{i=0}^{n} \mu_i^\phi h_i(\rho) \right) - \phi(\rho) \right\|_{L^2(\mathbb{R}, \gamma)} = 0. \tag{30}$$

This readily implies the following result.

**Proposition A.3.** *Let $\rho_1, \rho_2$ be two standard Gaussian random variables with correlation $\rho \in [-1, 1]$, and let $\phi, \tau \in L^2(\mathbb{R}, \gamma)$. Then,*

$$\mathbb{E}_{\rho_1, \rho_2} \left[ \phi(\rho_1) \tau(\rho_2) \right] = \sum_{i \in \mathbb{N}} \mu_i^\phi \mu_i^\tau \rho^i. \tag{31}$$

## B    PROOF OF LEMMA 4.1

We start by refreshing some useful notions of linear algebra. Let $A \in \mathbb{R}^{N \times p}$ be a matrix, with $p \geq N$, and $A_{-1} \in \mathbb{R}^{(N-1) \times p}$ be obtained from $A$ after removing the first row. We assume $AA^\top$ to be invertible, *i.e.*, the rows of $A$ are linearly independent. Thus, also the rows of $A_{-1}$ are linearly independent, implying that $A_{-1}A_{-1}^\top$ is invertible as well. We indicate with $P_A \in \mathbb{R}^{p \times p}$ the projector over $\text{Span}\{\text{rows}(A)\}$, and we correspondingly define $P_{A_{-1}} \in \mathbb{R}^{p \times p}$. As $AA^\top$ is invertible, we have that $\text{rank}(A) = N$.

By singular value decomposition, we have $A = UDO^\top$, where $U \in \mathbb{R}^{N \times N}$ and $O \in \mathbb{R}^{p \times p}$ are orthogonal matrices, and $D \in \mathbb{R}^{N \times p}$ contains the (all strictly positive) singular values of $A$ in its "left" diagonal, and is 0 in every other entry. Let us define $O_1 \in \mathbb{R}^{N \times p}$ as the matrix containing the first $N$ rows of $O$. This notation implies that if $O_1 u = 0$ for $u \in \mathbb{R}^p$, then $Au = 0$, *i.e.*, $u \in \text{Span}\{\text{rows}(A)\}^\perp$. The opposite implication is also true, which implies that $\text{Span}\{\text{rows}(A)\} = \text{Span}\{\text{rows}(O_1)\}$. As the rows of $O_1$ are orthogonal, we can then write

$$P_A = O_1^\top O_1. \tag{32}$$

We define $D_s \in \mathbb{R}^{N \times N}$, as the square, diagonal, and invertible matrix corresponding to the first $N$ columns of $D$. Let's also define $I_N \in \mathbb{R}^{p \times p}$ as the matrix containing 1 in the first $N$ entries of its diagonal, and 0 everywhere else. We have

$$\begin{aligned} P_A &= O_1^\top O_1 = OI_N O^\top \\ &= OD^\top D_s^{-2} DO^\top = OD^\top U^\top U D_s^{-2} U^\top U DO^\top \\ &= A^\top \left( UD_s^2 U^\top \right)^{-1} A = A^\top \left( UDO^\top OD^\top U^\top \right)^{-1} A \\ &= A^\top \left( AA^\top \right)^{-1} A \equiv A^+ A, \end{aligned} \tag{33}$$

where $A^+$ denotes the Moore-Penrose inverse.

Notice that this last form enables us to easily derive

$$P_{A_{-1}} A^+ v = A_{-1}^+ A_{-1} A^+ v = A_{-1}^+ I_{-1} AA^+ v = A_{-1}^+ I_{-1} v = A_{-1}^+ v_{-1}, \tag{34}$$

where $v \in \mathbb{R}^N$, $I_{-1} \in \mathbb{R}^{(N-1) \times N}$ is the $N \times N$ identity matrix without the first row, and $v_{-1} \in \mathbb{R}^{N-1}$ corresponds to $v$ without its first entry.

**Lemma B.1.** *Let $\Phi \in \mathbb{R}^{N \times k}$ be a matrix whose first row is denoted as $\varphi(z_1)$. Let $\Phi_{-1} \in \mathbb{R}^{(N-1) \times k}$ be the original matrix without the first row, and let $P_{\Phi_{-1}}$ be the projector over the* span *of its rows. Then,*

$$\left\| P_{\Phi_{-1}}^\perp \varphi(z_1) \right\|_2^2 \geq \lambda_{\min} \left( \Phi \Phi^\top \right). \tag{35}$$

*Proof.* If $\lambda_{\min} \left( \Phi \Phi^\top \right) = 0$, the thesis becomes trivial. Otherwise, we have that $\Phi \Phi^\top$, and therefore $\Phi_{-1} \Phi_{-1}^\top$, are invertible.

Let $u \in \mathbb{R}^N$ be a vector, such that its first entry $u_1 = 1$. We denote with $u_{-1} \in \mathbb{R}^{N-1}$ the vector $u$ without its first component, i.e. $u = [1, u_{-1}]$. We have

$$\left\| \Phi^\top u \right\|_2^2 \geq \lambda_{\min} \left( \Phi \Phi^\top \right) \|u\|_2^2 \geq \lambda_{\min} \left( \Phi \Phi^\top \right). \tag{36}$$

Setting $u_{-1} = - \left( \Phi_{-1} \Phi_{-1}^\top \right)^{-1} \Phi_{-1} \varphi(z_1)$, we get

$$\Phi^\top u = \varphi(z_1) + \Phi_{-1}^\top u_{-1} = \varphi(z_1) - P_{\Phi_{-1}} \varphi(z_1) = P_{\Phi_{-1}}^\perp \varphi(z_1). \tag{37}$$

Plugging this in (36), we get the thesis. $\square$

At this point, we are ready to prove Lemma 4.1.

*Proof of Lemma 4.1.* We indicate with $\Phi_{-1} \in \mathbb{R}^{(N-1) \times p}$ the feature matrix of the training set $\Phi \in \mathbb{R}^{N \times p}$ *without* the first sample $z_1$. In other words, $\Phi_{-1}$ is equivalent to $\Phi$, without the first row. Notice that since $K = \Phi \Phi^\top$ is invertible, also $K_{-1} := \Phi_{-1} \Phi_{-1}^\top$ is.

We can express the projector over the span of the rows of $\Phi$ in terms of the projector over the span of the rows of $\Phi_{-1}$ as follows

$$P_\Phi = P_{\Phi_{-1}} + \frac{P_{\Phi_{-1}}^\perp \varphi(z_1) \varphi(z_1)^\top P_{\Phi_{-1}}^\perp}{\left\| P_{\Phi_{-1}}^\perp \varphi(z_1) \right\|_2^2}. \tag{38}$$

The above expression is a consequence of the Gram-Schmidt formula, and the quantity at the denominator is different from zero because of Lemma B.1, as $K$ is invertible.

We indicate with $\Phi^+ = \Phi^\top K^{-1}$ the Moore–Penrose pseudo-inverse of $\Phi$. Using (3), we can define $\theta_{-1}^* := \theta_0 + \Phi_{-1}^+ \left( G_{-1} - f(Z_{-1}, \theta_0) \right)$, *i.e.*, the set of parameters the algorithm would have converged to if trained over $(Z_{-1}, G_{-1})$, the original data-set without the first pair sample-label $(z_1, g_1)$.

Notice that $P_\Phi \Phi^\top = \Phi^\top$, as a consequence of (33). Thus, again using (3), for any $z$ we can write

$$\begin{aligned} f(z, \theta^*) - \varphi(z)^\top \theta_0 &= \varphi(z)^\top \Phi^+ \left( G - f(Z, \theta_0) \right) \\ &= \varphi(z)^\top P_\Phi \Phi^+ \left( G - f(Z, \theta_0) \right) \\ &= \varphi(z)^\top \left( P_{\Phi_{-1}} + \frac{P_{\Phi_{-1}}^\perp \varphi(z_1) \varphi(z_1)^\top P_{\Phi_{-1}}^\perp}{\left\| P_{\Phi_{-1}}^\perp \varphi(z_1) \right\|_2^2} \right) \Phi^+ \left( G - f(Z, \theta_0) \right). \end{aligned} \tag{39}$$

Notice that, thanks to (34), we can manipulate the first term in the bracket as follows

$$\begin{aligned} \varphi(z)^\top P_{\Phi_{-1}} \Phi^+ \left( G - f(Z, \theta_0) \right) &= \varphi(z)^\top \Phi_{-1}^+ \left( G_{-1} - f(Z_{-1}, \theta_0) \right) \\ &= f(z, \theta_{-1}^*) - \varphi(z)^\top \theta_0. \end{aligned} \tag{40}$$

Thus, bringing the result of (40) on the LHS, (39) becomes

$$\begin{aligned} f(z, \theta^*) - f(z, \theta_{-1}^*) &= \frac{\varphi(z)^\top P_{\Phi_{-1}}^\perp \varphi(z_1)}{\left\| P_{\Phi_{-1}}^\perp \varphi(z_1) \right\|_2^2} \varphi(z_1)^\top P_{\Phi_{-1}}^\perp \Phi^+ \left( G - f(Z, \theta_0) \right) \\ &= \frac{\varphi(z)^\top P_{\Phi_{-1}}^\perp \varphi(z_1)}{\left\| P_{\Phi_{-1}}^\perp \varphi(z_1) \right\|_2^2} \varphi(z_1)^\top \left( I - P_{\Phi_{-1}} \right) \Phi^+ \left( G - f(Z, \theta_0) \right) \\ &= \frac{\varphi(z)^\top P_{\Phi_{-1}}^\perp \varphi(z_1)}{\left\| P_{\Phi_{-1}}^\perp \varphi(z_1) \right\|_2^2} \left( f(z_1, \theta^*) - f(z_1, \theta_{-1}^*) \right), \end{aligned} \tag{41}$$

where in the last step we again used (3) and (40). $\square$

## C  USEFUL LEMMAS

**Lemma C.1.** *Let $x$ and $y$ be two Lipschitz concentrated, independent random vectors. Let $\zeta(x,y)$ be a Lipschitz function in both arguments, i.e., for every $\delta$,*

$$
\begin{aligned}
|\zeta(x+\delta,y) - \zeta(x,y)| &\leq L \, \|\delta\|_2, \\
|\zeta(x,y+\delta) - \zeta(x,y)| &\leq L \, \|\delta\|_2,
\end{aligned}
\tag{42}
$$

*for all $x$ and $y$. Then, $\zeta(x,y)$ is a Lipschitz concentrated random variable, in the joint probability space of $x$ and $y$.*

*Proof.* To prove the thesis, we need to show that, for every 1-Lipschitz function $\tau$, the following holds

$$
\mathbb{P}_{xy}\left(|\tau\left(\zeta(x,y)\right) - \mathbb{E}_{xy}\left[\tau\left(\zeta(x,y)\right)\right]| > t\right) < e^{-ct^2},
\tag{43}
$$

where $c$ is a universal constant. An application of the triangle inequality gives

$$
\begin{aligned}
&|\tau\left(\zeta(x,y)\right) - \mathbb{E}_{xy}\left[\tau\left(\zeta(x,y)\right)\right]| \\
&\leq |\tau\left(\zeta(x,y)\right) - \mathbb{E}_x\left[\tau\left(\zeta(x,y)\right)\right]| + |\mathbb{E}_x\left[\tau\left(\zeta(x,y)\right)\right] - \mathbb{E}_y\mathbb{E}_x\left[\tau\left(\zeta(x,y)\right)\right]| =: A + B.
\end{aligned}
\tag{44}
$$

Thus, we can upper bound LHS of (43) as follows:

$$
\mathbb{P}_{xy}\left(|\tau\left(\zeta(x,y)\right) - \mathbb{E}_{xy}\left[\tau\left(\zeta(x,y)\right)\right]| > t\right) \leq \mathbb{P}_{xy}\left(A + B > t\right).
\tag{45}
$$

If $A$ and $B$ are positive random variables, it holds that $\mathbb{P}(A + B > t) \leq \mathbb{P}(A > t/2) + \mathbb{P}(B > t/2)$. Then, the LHS of (43) is also upper bounded by

$$
\begin{aligned}
&\mathbb{P}_{xy}\left(|\tau\left(\zeta(x,y)\right) - \mathbb{E}_x\left[\tau\left(\zeta(x,y)\right)\right]| > t/2\right) \\
&+ \mathbb{P}_{xy}\left(|\mathbb{E}_x\left[\tau\left(\zeta(x,y)\right)\right] - \mathbb{E}_y\mathbb{E}_x\left[\tau\left(\zeta(x,y)\right)\right]| > t/2\right).
\end{aligned}
\tag{46}
$$

Since $\tau \circ \zeta$ is Lipschitz with respect to $x$ for every $y$, we have

$$
\mathbb{P}_{xy}\left(|\tau\left(\zeta(x,y)\right) - \mathbb{E}_x\left[\tau\left(\zeta(x,y)\right)\right]| > t/2\right) < e^{-c_1 t^2},
\tag{47}
$$

for some absolute constant $c_1$. Furthermore, $\chi(y) := \mathbb{E}_x\left[\tau\left(\zeta(x,y)\right)\right]$ is also Lipschitz, as

$$
\begin{aligned}
|\chi(y+\delta) - \chi(y)| &= |\mathbb{E}_x\left[\tau\left(\zeta(x,y+\delta)\right) - \tau\left(\zeta(x,y)\right)\right]| \\
&\leq \mathbb{E}_x\left[|\tau\left(\zeta(x,y+\delta)\right) - \tau\left(\zeta(x,y)\right)|\right] \leq L\,\|\delta\|_2.
\end{aligned}
\tag{48}
$$

Then, we can write

$$
\begin{aligned}
&\mathbb{P}_{xy}\left(|\mathbb{E}_x\left[\tau\left(\zeta(x,y)\right)\right] - \mathbb{E}_y\mathbb{E}_x\left[\tau\left(\zeta(x,y)\right)\right]| > t/2\right) \\
&= \mathbb{P}_y\left(|\chi(y) - \mathbb{E}_y\left[\chi(y)\right]| > t/2\right) < e^{-c_2 t^2},
\end{aligned}
\tag{49}
$$

for some absolute constant $c_2$. Thus,

$$
\mathbb{P}_{xy}\left(|\tau\left(\zeta(x,y)\right) - \mathbb{E}_{xy}\left[\tau\left(\zeta(x,y)\right)\right]| > t\right) < e^{-c_1 t^2} + e^{-c_2 t^2} \leq e^{-ct^2},
\tag{50}
$$

for some absolute constant $c$, which concludes the proof. $\qquad\square$

**Lemma C.2.** *Let $x \sim \mathcal{P}_X$, $y \sim \mathcal{P}_Y$ and $z = [x,y] \sim \mathcal{P}_Z$. Let Assumption 5.1 hold. Then, $z$ is a Lipschitz concentrated random vector.*

*Proof.* We want to prove that, for every 1-Lipschitz function $\tau$, the following holds

$$
\mathbb{P}_z\left(|\tau\left(z\right) - \mathbb{E}_z\left[\tau\left(z\right)\right]| > t\right) < e^{-ct^2},
\tag{51}
$$

for some universal constant $c$. As we can write $z = [x,y]$, defining $z' = [x',y]$, we have

$$
|\tau\left(z\right) - \tau\left(z'\right)| \leq \|z - z'\|_2 = \|x - x'\|_2,
\tag{52}
$$

*i.e.*, for every $y$, $\tau$ is 1-Lipschitz with respect to $x$. The same can be shown for $y$, with an equivalent argument. Since $x$ and $y$ are independent random vectors, both Lipschitz concentrated, Lemma C.1 gives the thesis. $\qquad\square$

**Lemma C.3.** *Let $\tau$ and $\zeta$ be two Lipschitz functions. Let $z, z' \in \mathbb{R}^d$ be two fixed vectors such that $\|z\|_2 = \|z'\|_2 = \sqrt{d}$. Let $V$ be a $k \times d$ matrix such that $V_{i,j} \sim_{\text{i.i.d.}} \mathcal{N}(0, 1/d)$. Then, for any $t > 1$,*

$$\left| \tau(Vz)^\top \zeta(Vz') - \mathbb{E}_V \left[ \tau(Vz)^\top \zeta(Vz') \right] \right| = \mathcal{O}\left( \sqrt{k} \log t \right), \tag{53}$$

*with probability at least $1 - \exp(-c \log^2 t)$ over $V$. Here, $\tau$ and $\zeta$ act component-wise on their arguments. Furthermore, by taking $\tau = \zeta$ and $z = z'$, we have that*

$$\mathbb{E}_V \left[ \|\tau(Vz)\|_2^2 \right] = k \mathbb{E}_\rho \left[ \tau^2(\rho) \right], \tag{54}$$

*where $\rho \sim \mathcal{N}(0, 1)$. This implies that $\|\tau(Vz)\|_2^2 = \mathcal{O}(k)$ with probability at least $1 - \exp(-ck)$ over $V$.*

*Proof.* We have

$$\tau(Vz)^\top \zeta(Vz') = \sum_{j=1}^k \tau(v_j^\top z) \zeta(v_j^\top z'), \tag{55}$$

where we used the shorthand $v_j := V_{j:}$. As $\tau$ and $\zeta$ are Lipschitz, $v_j \sim \mathcal{N}(0, I/d)$, and $\|z\|_2 = \|z'\|_2 = \sqrt{d}$, we have that $\tau(Vz)^\top \zeta(Vz')$ is the sum of $k$ independent sub-exponential random variables, in the probability space of $V$. Thus, by Bernstein inequality (cf. Theorem 2.8.1 in Vershynin (2018)), we have

$$\left| \tau(Vz)^\top \zeta(Vz') - \mathbb{E}_V \left[ \tau(Vz)^\top \zeta(Vz') \right] \right| = \mathcal{O}\left( \sqrt{k} \log t \right). \tag{56}$$

with probability at least $1 - \exp(-c \log^2 t)$, over the probability space of $V$, which gives the thesis. The second statement is again implied by the fact that $v_j \sim \mathcal{N}(0, I/d)$ and $\|z\|_2 = \sqrt{d}$. $\qquad\square$

**Lemma C.4.** *Let $x, x_1 \sim \mathcal{P}_X$ and $y_1 \sim \mathcal{P}_Y$ be independent random variables, with $x, x_1 \in \mathbb{R}^{d_x}$ and $y_1 \in \mathbb{R}^{d_y}$, and let Assumption 5.1 hold. Let $d = d_x + d_y$, $V$ be a $k \times d$ matrix, such that $V_{i,j} \sim_{\text{i.i.d.}} \mathcal{N}(0, 1/d)$, and let $\tau$ be a Lipschitz function. Let $z_1 := [x_1, y_1]$ and $z_1^m := [x, y_1]$. Let $\alpha = d_y/d \in (0, 1)$ and $\mu_l$ be the $l$-th Hermite coefficient of $\tau$. Then, for any $t > 1$,*

$$\left| \tau(Vz_1^m)^\top \tau(Vz_1) - k \sum_{l=0}^{+\infty} \mu_l^2 \alpha^l \right| = \mathcal{O}\left( \sqrt{k} \left( \sqrt{\frac{k}{d}} + 1 \right) \log t \right), \tag{57}$$

*with probability at least $1 - \exp(-c \log^2 t) - \exp(-ck)$ over $V$ and $x$, where $c$ is a universal constant.*

*Proof.* Define the vector $x'$ as follows

$$x' = \frac{\sqrt{d_x} \left( I - \frac{x_1 x_1^\top}{d_x} \right) x}{\left\| \left( I - \frac{x_1 x_1^\top}{d_x} \right) x \right\|_2}. \tag{58}$$

Note that, by construction, $x_1^\top x' = 0$ and $\|x'\|_2 = \sqrt{d_x}$. Also, consider a vector $y$ orthogonal to both $x_1$ and $x$. Then, a fast computation returns $y^\top x' = 0$. This means that $x'$ is the vector on the $\sqrt{d_x}$-sphere, lying on the same plane of $x_1$ and $x$, orthogonal to $x_1$. Thus, we can easily compute

$$\frac{|x^\top x'|}{d_x} = \sqrt{1 - \left( \frac{x^\top x_1}{d_x} \right)^2} \geq 1 - \left( \frac{x^\top x_1}{d_x} \right)^2, \tag{59}$$

where the last inequality derives from $\sqrt{1 - a} \geq 1 - a$ for $a \in [0, 1]$. Then,

$$\|x - x'\|_2^2 = \|x\|_2^2 + \|x'\|_2^2 - 2x^\top x' \leq 2d_x \left( 1 - \left( 1 - \left( \frac{x^\top x_1}{d_x} \right)^2 \right) \right) = 2 \frac{(x^\top x_1)^2}{d_x}. \tag{60}$$

As $x$ and $x_1$ are both sub-Gaussian, mean-0 vectors, with $\ell_2$ norm equal to $\sqrt{d_x}$, we have that

$$\mathbb{P}\left(\|x - x'\|_2 > t\right) \leq \mathbb{P}\left(|x^\top x_1| > \sqrt{d_x} t / \sqrt{2}\right) < \exp(-ct^2), \tag{61}$$

where $c$ is an absolute constant. Here the probability is referred to the space of $x$, for a fixed $x_1$. Thus, $\|x - x'\|_2$ is sub-Gaussian.

We now define $z' := [x', y_1]$. Notice that $z_1^\top z' = \|y_1\|_2^2 = d_y$ and $\|z_1^m - z'\|_2 = \|x - x'\|_2$. We can write

$$\begin{aligned}
\left|\tau(V z_1^m)^\top \tau(V z_1) - \tau(V z')^\top \tau(V z_1)\right| &\leq \|\tau(V z_1^m) - \tau(V z')\|_2 \|\tau(V z_1)\|_2 \\
&\leq C \|V\|_{\mathrm{op}} \|z_1^m - z'\|_2 \|\tau(V z_1)\|_2 \\
&\leq C_1 \left(\sqrt{\frac{k}{d}} + 1\right) \|x - x'\|_2 \sqrt{k} \\
&= \mathcal{O}\left(\sqrt{k}\left(\sqrt{\frac{k}{d}} + 1\right) \log t\right).
\end{aligned} \tag{62}$$

Here the second step holds as $\tau$ is Lipschitz; the third step holds with probability at least $1 - \exp(-c_1 \log^2 t) - \exp(-c_2 k)$, and it uses Theorem 4.4.5 of Vershynin (2018) and Lemma C.3; the fourth step holds with probability at least $1 - \exp(-c \log^2 t)$, and it uses (61). This probability is intended over $V$ and $x$. We further have

$$\left|\tau(V z')^\top \tau(V z_1) - \mathbb{E}_V\left[\tau(V z')^\top \tau(V z_1)\right]\right| = \mathcal{O}\left(\sqrt{k} \log t\right), \tag{63}$$

with probability at least $1 - \exp(-c_3 \log^2 t) - \exp(-c_2 k)$ over $V$, because of Lemma C.3.

We have

$$\mathbb{E}_V\left[\tau(V z')^\top \tau(V z_1)\right] = k \mathbb{E}_{\rho_1 \rho_2}\left[\tau(\rho_1)\tau(\rho_2)\right], \tag{64}$$

where we indicate with $\rho_1$ and $\rho_2$ two standard Gaussian random variables, with correlation

$$\mathrm{corr}(\rho_1, \rho_2) = \frac{z_1^\top z}{\|z_1\|_2 \|z'\|_2} = \frac{d_y}{d} = \alpha. \tag{65}$$

Then, exploiting the Hermite expansion of $\tau$, we have

$$\mathbb{E}_{\rho_1 \rho_2}\left[\tau(\rho_1)\tau(\rho_2)\right] = \sum_{l=0}^{+\infty} \mu_l^2 \alpha^l. \tag{66}$$

Putting together (62), (63), (64), and (66) gives the thesis. $\qquad\square$

## D    Proofs for Random Features

In this section, we indicate with $Z \in \mathbb{R}^{N \times d}$ the data matrix, such that its rows are sampled independently from $\mathcal{P}_Z$ (see Assumption 5.1). We denote by $V \in \mathbb{R}^{k \times d}$ the random features matrix, such that $V_{ij} \sim_{\mathrm{i.i.d.}} \mathcal{N}(0, 1/d)$. Thus, the feature map is given by (see (14))

$$\varphi(z) := \phi(V z) \in \mathbb{R}^k, \tag{67}$$

where $\phi$ is the activation function, applied component-wise to the pre-activations $V z$. We use the shorthands $\Phi := \phi(Z V^\top) \in \mathbb{R}^{N \times k}$ and $K := \Phi \Phi^\top \in \mathbb{R}^{N \times N}$, we indicate with $\Phi_{-1} \in \mathbb{R}^{(N-1) \times k}$ the matrix $\Phi$ without the first row, and we define $K_{-1} := \Phi_{-1} \Phi_{-1}^\top$. We call $P_\Phi$ the projector over the span of the rows of $\Phi$, and $P_{\Phi_{-1}}$ the projector over the span of the rows of $\Phi_{-1}$. We use the notations $\tilde{\varphi}(z) := \varphi(z) - \mathbb{E}_V[\varphi(z)]$ and $\tilde{\Phi}_{-1} := \Phi_{-1} - \mathbb{E}_V[\Phi_{-1}]$ to indicate the centered feature map and matrix respectively, where the centering is with respect to $V$. We indicate with $\mu_l$ the $l$-th Hermite coefficient of $\phi$. We use the notation $z_1^m = [x, y_1]$, where $x \sim \mathcal{P}_X$ is sampled independently from $V$ and $Z$. We denote by $V_x$ ($V_y$) the first $d_x$ (last $d_y$) columns of $V$, i.e., $V = [V_x, V_y]$. We define $\alpha = d_y/d$. Throughout this section, for compactness, we drop the subscripts "RF" from these quantities, as we will only treat the proofs related to Section 5. Again for the sake of compactness, we will not re-introduce such quantities in the statements or the proofs of the following lemmas.

The content of this section can be summarized as follows:

- In Lemma D.2 we prove a lower bound on the smallest eigenvalue of $K$, adapting to our settings Lemma C.5 of Bombari et al. (2023). As our assumptions are less restrictive than those in Bombari et al. (2023), we will crucially exploit Lemma D.1.

- In Lemma D.3, we treat separately a term that derives from $\mathbb{E}_V [\phi(Vz)] = \mu_0 \mathbf{1}_k$, showing that we can *center* the activation function, without changing our final statement in Theorem 5.4. This step is necessary only if $\mu_0 \neq 0$.

- In Lemma D.4, we show that the non-linear component of the features $\tilde{\varphi}(z_1) - \mu_1 V z_1$ and $\tilde{\varphi}(z_1^m) - \mu_1 V z_1^m$ have a negligible component in the space spanned by the rows of $\Phi_{-1}$.

- In Lemma D.7, we provide concentration results for $\varphi(z_1^m)^\top P_{\Phi_{-1}}^\perp \varphi(z_1)$, and we lower bound this same term in Lemma D.6, exploiting also the intermediate result provided in Lemma D.5.

- Finally, we prove Theorem 5.4.

**Lemma D.1.** *Let $A := (Z^{*m}) \in \mathbb{R}^{N \times d^m}$, for some natural $m \geq 2$, where $*$ refers to the Khatri-Rao product, defined in Appendix A. We have*

$$\lambda_{\min} \left( AA^\top \right) = \Omega(d^m), \tag{68}$$

*with probability at least $1 - \exp(-c \log^2 N)$ over $Z$, where $c$ is an absolute constant.*

*Proof.* As $m \geq 2$, we can write $A = \left( Z^{*2} \right) * \left( Z^{*(m-2)} \right) =: A_2 * A_m$ (where $\left( Z^{*0} \right)$ is defined to be the vector full of ones $\mathbf{1}_N \in \mathbb{R}^N$). We can provide a lower bound on the smallest eigenvalue of such product through the following inequality Schur (1911):

$$\lambda_{\min} \left( AA^\top \right) = \lambda_{\min} \left( A_2 A_2^\top \circ A_m A_m^\top \right) \geq \lambda_{\min} \left( A_2 A_2^\top \right) \min_i \|(A_m)_{i:}\|_2^2. \tag{69}$$

Note that the rows of $Z$ are mean-0 and Lipschitz concentrated by Lemma C.2. Then, by following the argument of Lemma C.3 in Bombari et al. (2023), we have

$$\lambda_{\min} \left( A_2 A_2^\top \right) = \Omega(d^2), \tag{70}$$

with probability at least $1 - \exp(-c \log^2 N)$ over $Z$. We remark that, for the argument of Lemma C.3 in Bombari et al. (2023) to go through, it suffices that $N = o(d^2 / \log^4 d)$ and $N \log^4 N = o(d^2)$ (see Equations (C.23) and (C.26) in Bombari et al. (2023)), which is implied by Assumption 5.2, despite it being milder than Assumption 4 in Bombari et al. (2023).

For the second term of (69), we have

$$\|(A_m)_{i:}\|_2^2 = \|z_i\|_2^{2(m-2)} = d^{m-2}, \tag{71}$$

due to Assumption 5.1. Thus, the thesis readily follows. $\square$

**Lemma D.2.** *We have that*

$$\lambda_{\min} (K) = \Omega(k), \tag{72}$$

*with probability at least $1 - \exp \left( -c \log^2 N \right)$ over $V$ and $Z$, where $c$ is an absolute constant. This implies that $\lambda_{\min} (K_{-1}) = \Omega(k)$.*

*Proof.* The proof follows the same path as Lemma C.5 of Bombari et al. (2023). In particular, we define a truncated version of $\Phi$ as follows

$$\bar{\Phi}_{:j} = \phi(Zv_j)\chi \left( \|\phi(Zv_j)\|_2^2 \leq R \right), \tag{73}$$

where $\chi$ is the indicator function and we introduce the shorthand $v_i := V_{i:}$. In this case, $\chi = 1$ if $\|\phi(Zv_j)\|_2^2 \leq R$, and $\chi = 0$ otherwise. As this is a column-wise truncation, it's easy to verify that $\Phi\Phi^\top \succeq \bar{\Phi}\bar{\Phi}^\top$. Over such truncated matrix, we can use Matrix Chernoff inequality (see Theorem 1.1 of Tropp (2012)), which gives that $\lambda_{\min} \left( \bar{\Phi}\bar{\Phi}^\top \right) = \Omega(\lambda_{\min} \left( \bar{G} \right))$, where $\bar{G} := \mathbb{E}_V \left[ \bar{\Phi}\bar{\Phi}^\top \right]$. Finally, we prove closeness between $\bar{G}$ and $G$, which is analogously defined as $G := \mathbb{E}_V \left[ \Phi\Phi^\top \right]$.

To be more specific, setting $R = k/\log^2 N$, we have

$$\lambda_{\min}(K) \geq \lambda_{\min}(\bar{\Phi}\bar{\Phi}^\top) \geq \lambda_{\min}(\bar{G})/2 \geq \lambda_{\min}(G)/2 - o(k), \tag{74}$$

where the second inequality holds with probability at least $1 - \exp(c_1 \log^2 N)$ over $V$, if $\lambda_{\min}(G) = \Omega(k)$ (see Equation (C.47) of Bombari et al. (2023)), and the third comes from Equation (C.45) in Bombari et al. (2023). To perform these steps, our Assumptions 5.2 and 5.3 are enough, despite the second one being milder than Assumption 2 in Bombari et al. (2023).

To conclude the proof, we are left to prove that $\lambda_{\min}(G) = \Omega(k)$ with probability at least $1 - \exp(-c_2 \log^2 N)$ over $V$ and $Z$.

We have that

$$G = \mathbb{E}_V[K] = \mathbb{E}_V\left[\sum_{i=1}^{k} \phi(ZV_{i:}^\top)\phi(ZV_{i:}^\top)^\top\right] = k\mathbb{E}_v\left[\phi(Zv)\phi(Zv)^\top\right] := kM, \tag{75}$$

where we use the shorthand $v$ to indicate a random variable distributed as $V_{1:}$. We also indicate with $z_i$ the $i$-th row of $Z$. Exploiting the Hermite expansion of $\phi$, we can write

$$M_{ij} = \mathbb{E}_v\left[\phi(z_i^\top v)\phi(z_j^\top v)\right] = \sum_{l=0}^{+\infty} \mu_l^2 \frac{(z_i^\top z_j)^l}{d^l} = \sum_{l=0}^{+\infty} \mu_l^2 \frac{\left[(Z^{*l})(Z^{*l})^\top\right]_{ij}}{d^l}, \tag{76}$$

where $\mu_l$ is the $l$-th Hermite coefficient of $\phi$. Note that the previous expansion was possible since $\|z_i\| = \sqrt{d}$ for all $i \in [N]$. As $\phi$ is non-linear, there exists $m \geq 2$ such that $\mu_m^2 > 0$. In particular, we have $M \succeq \frac{\mu_m^2}{d^m} AA^\top$ in a PSD sense, where we define

$$A := (Z^{*m}). \tag{77}$$

By Lemma D.1, the desired result readily follows. $\qquad\square$

**Lemma D.3.** *Let $\mu_0 \neq 0$. Then,*

$$\left\|P_{\Phi_{-1}}^\perp \mathbf{1}_k\right\|_2 = o(\sqrt{k}), \tag{78}$$

*with probability at least $1 - e^{-cd} - e^{-cN}$ over $V$ and $Z$, where $c$ is an absolute constant.*

*Proof.* Note that $\Phi_{-1}^\top = \mu_0 \mathbf{1}_k \mathbf{1}_{N-1}^\top + \tilde{\Phi}_{-1}^\top$. Here, $\tilde{\Phi}_{-1}^\top$ is a $k \times (N-1)$ matrix with i.i.d. and mean-0 rows, whose sub-Gaussian norm (in the probability space of $V$) can be bounded as

$$\left\|\tilde{\Phi}_{:i}\right\|_{\psi_2} = \|\phi(ZV_{i:}) - \mathbb{E}_V[\phi(ZV_{i:})]\|_{\psi_2} \leq L\frac{\|Z\|_{\text{op}}}{\sqrt{d}} = \mathcal{O}\left(\sqrt{N/d} + 1\right), \tag{79}$$

where first inequality holds since $\phi$ is $L$-Lipschitz and $V_{i:}$ is a Gaussian (and hence, Lipschitz concentrated) vector with covariance $I/d$. The last step holds with probability at least $1 - e^{-cd}$ over $Z$, because of Lemma B.7 in Bombari et al. (2022b).

Thus, another application of Lemma B.7 in Bombari et al. (2022b) gives

$$\left\|\tilde{\Phi}_{-1}^\top\right\|_{\text{op}} = \mathcal{O}\left(\left(\sqrt{k} + \sqrt{N}\right)\left(\sqrt{N/d} + 1\right)\right) = \mathcal{O}\left(\sqrt{k}\left(\sqrt{N/d} + 1\right)\right), \tag{80}$$

where the first equality holds with probability at least $1 - e^{-cN}$ over $V$, and the second is a direct consequence of Assumption 5.2.

We can write

$$\Phi_{-1}^\top \frac{\mathbf{1}_{N-1}}{\mu_0(N-1)} = \left(\mu_0 \mathbf{1}_k \mathbf{1}_{N-1}^\top + \tilde{\Phi}_{-1}^\top\right)\frac{\mathbf{1}_{N-1}}{\mu_0(N-1)} = \mathbf{1}_k + \tilde{\Phi}_{-1}^\top \frac{\mathbf{1}_{N-1}}{\mu_0(N-1)} =: \mathbf{1}_k + v, \tag{81}$$

where

$$\|v\|_2 \leq \frac{1}{\mu_0(N-1)}\left\|\tilde{\Phi}_{-1}^\top\right\|_{\text{op}}\|\mathbf{1}_{N-1}\|_2 = \mathcal{O}\left(\sqrt{\frac{k}{N}}\left(\sqrt{N/d} + 1\right)\right) = o(\sqrt{k}). \tag{82}$$

Thus, we can conclude

$$\left\| P_{\Phi_{-1}}^{\perp} \mathbf{1}_k \right\|_2 = \left\| P_{\Phi_{-1}}^{\perp} \left( \Phi_{-1}^{\top} \frac{\mathbf{1}_{N-1}}{\mu_0(N-1)} - v \right) \right\|_2$$
$$\leq \left\| P_{\Phi_{-1}}^{\perp} P_{\Phi_{-1}} \Phi_{-1}^{\top} \frac{\mathbf{1}_{N-1}}{\mu_0(N-1)} \right\|_2 + \|v\|_2 = o(\sqrt{k}), \tag{83}$$

where in the second step we use the triangle inequality, $\Phi_{-1}^{\top} = P_{\Phi_{-1}} \Phi_{-1}^{\top}$, and $\left\| P_{\Phi_{-1}}^{\perp} v \right\|_2 \leq \|v\|_2$. $\qquad\square$

**Lemma D.4.** *Let $z \sim \mathcal{P}_Z$, sampled independently from $Z_{-1}$, and denote $\tilde{\phi}(x) := \phi(x) - \mu_0$. Then,*

$$\left\| P_{\Phi_{-1}} \left( \tilde{\phi}(Vz) - \mu_1 Vz \right) \right\|_2 = o(\sqrt{k}), \tag{84}$$

*with probability at least $1 - \exp\left(-c\log^2 N\right)$ over $V$, $Z_{-1}$ and $z$, where $c$ is an absolute constant.*

*Proof.* As $P_{\Phi_{-1}} = \Phi_{-1}^{+} \Phi_{-1}$, we have

$$\left\| P_{\Phi_{-1}} \left( \tilde{\phi}(Vz) - \mu_1 Vz \right) \right\|_2 \leq \left\| \Phi_{-1}^{+} \right\|_{\mathrm{op}} \left\| \Phi_{-1} \left( \tilde{\phi}(Vz) - \mu_1 Vz \right) \right\|_2$$
$$= \mathcal{O} \left( \frac{\left\| \Phi_{-1} \left( \tilde{\phi}(Vz) - \mu_1 Vz \right) \right\|_2}{\sqrt{k}} \right), \tag{85}$$

where the last equality holds with probability at least $1 - \exp\left(-c\log^2 N\right)$ over $V$ and $Z_{-1}$, because of Lemma D.2.

An application of Lemma C.3 with $t = N$ gives

$$|u_i - \mathbb{E}_V[u_i]| = \mathcal{O}\left( \sqrt{k} \log N \right), \tag{86}$$

where $u_i$ is the $i$-th entry of the vector $u := \Phi_{-1} \left( \tilde{\phi}(Vz) - \mu_1 Vz \right)$. This can be done since both $\phi$ and $\tilde{\phi} \equiv \phi - \mu_0$ are Lipschitz, $v_j \sim \mathcal{N}(0, I/d)$, and $\|z\|_2 = \|z_{i+1}\|_2 = \sqrt{d}$. Performing a union bound over all entries of $u$, we can guarantee that the previous equation holds for every $1 \leq i \leq N-1$, with probability at least $1 - (N-1)\exp(-c\log^2 N) \geq 1 - \exp(-c_1 \log^2 N)$. Thus, we have

$$\|u - \mathbb{E}_V[u]\|_2 = \mathcal{O}\left( \sqrt{k}\sqrt{N} \log N \right) = o(k), \tag{87}$$

where the last equality holds because of Assumption 5.2.

Note that the function $f(x) := \tilde{\phi}(x) - \mu_1 x$ has the first 2 Hermite coefficients equal to 0. Hence, as $v_i^{\top} z$ and $v_i^{\top} z_i$ are standard Gaussian random variables with correlation $\frac{z^{\top} z_i}{\|z\|_2 \|z_i\|_2}$, we have

$$|\mathbb{E}_V[u_i]| \leq k \sum_{l=2}^{+\infty} \mu_l^2 \left( \frac{|z^{\top} z_i|}{\|z\|_2 \|z_i\|_2} \right)^l$$
$$\leq k \max_l \mu_l^2 \sum_{l=2}^{+\infty} \left( \frac{|z^{\top} z_i|}{\|z\|_2 \|z_i\|_2} \right)^l$$
$$= k \max_l \mu_l^2 \left( \frac{z^{\top} z_i}{\|z\|_2 \|z_i\|_2} \right)^2 \frac{1}{1 - \frac{|z^{\top} z_i|}{\|z\|_2 \|z_i\|_2}} \tag{88}$$
$$\leq 2k \max_l \mu_l^2 \left( \frac{z^{\top} z_i}{\|z\|_2 \|z_i\|_2} \right)^2 = \mathcal{O}\left( \frac{k \log^2 N}{d} \right),$$

where the last inequality holds with probability at least $1 - \exp\left(-c\log^2 N\right)$ over $z$ and $z_i$, as they are two independent, mean-0, sub-Gaussian random vectors. Again, performing a union bound over

all entries of $\mathbb{E}_V[u]$, we can guarantee that the previous equation holds for every $1 \le i \le N-1$, with probability at least $1 - (N-1)\exp(-c\log^2 N) \ge 1 - \exp(-c_1 \log^2 N)$. Then, we have

$$\|\mathbb{E}_V[u]\|_2 = \mathcal{O}\left(\sqrt{N} \frac{k\log^2 N}{d}\right) = o(k), \tag{89}$$

where the last equality is a consequence of Assumption 5.2.

Finally, (87) and (89) give

$$\left\|\Phi_{-1}\left(\tilde{\phi}(Vz) - \mu_1 Vz\right)\right\|_2 \le \|\mathbb{E}_V[u]\|_2 + \|u - \mathbb{E}_V[u]\|_2 = o(k), \tag{90}$$

which plugged in (85) readily provides the thesis. $\qquad\square$

**Lemma D.5.** *We have*

$$\left|(Vz_1^m)^\top P_{\Phi_{-1}}^\perp Vz_1 - \left\|P_{\Phi_{-1}}^\perp V_y y_1\right\|_2^2\right| = o(k), \tag{91}$$

*with probability at least $1 - \exp(-c\log^2 N)$ over $x$, $z_1$ and $V$, where $c$ is an absolute constant.*

*Proof.* We have

$$Vz_1^m = V_x x + V_y y_1, \qquad Vz_1 = V_x x_1 + V_y y_1. \tag{92}$$

Thus, we can write

$$\left|(Vz_1^m)^\top P_{\Phi_{-1}}^\perp Vz_1 - \left\|P_{\Phi_{-1}}^\perp V_y y_1\right\|_2^2\right| = \left|(V_x x)^\top P_{\Phi_{-1}}^\perp Vz_1 + (V_y y_1)^\top P_{\Phi_{-1}}^\perp V_x x_1\right|$$
$$\le \left|x^\top V_x^\top P_{\Phi_{-1}}^\perp Vz_1\right| + \left|y_1^\top V_y^\top P_{\Phi_{-1}}^\perp V_x x_1\right|. \tag{93}$$

Let's look at the first term of the RHS of the previous equation. Notice that $\|V\|_{\mathrm{op}} = \mathcal{O}\left(\sqrt{k/d} + 1\right)$ with probability at least $1 - 2e^{-cd}$, because of Theorem 4.4.5 of Vershynin (2018). We condition on such event until the end of the proof, which also implies having the same bound on $\|V_x\|_{\mathrm{op}}$ and $\|V_y\|_{\mathrm{op}}$. Since $x$ is a mean-0 sub-Gaussian vector, independent from $V_x^\top P_{\Phi_{-1}}^\perp Vz_1$, we have

$$\left|x^\top V_x^\top P_{\Phi_{-1}}^\perp Vz_1\right| \le \log N \left\|V_x^\top P_{\Phi_{-1}}^\perp Vz_1\right\|_2$$
$$\le \log N \|V_x\|_{\mathrm{op}} \left\|P_{\Phi_{-1}}^\perp\right\|_{\mathrm{op}} \|V\|_{\mathrm{op}} \|z_1\|$$
$$= \mathcal{O}\left(\log N \left(\frac{k}{d} + 1\right)\sqrt{d}\right) = o(k), \tag{94}$$

where the first inequality holds with probability at least $1 - \exp(-c\log^2 N)$ over $x$, and the last line holds because $\left\|P_{\Phi_{-1}}^\perp\right\|_{\mathrm{op}} \le 1$, $\|z_1\| = \sqrt{d}$, and because of Assumption 5.2.

Similarly, exploiting the independence between $x_1$ and $y_1$, we can prove that $\left|y_1^\top V_y^\top P_{\Phi_{-1}}^\perp V_x x_1\right| = o(k)$, with probability at least $1 - \exp(-c\log^2 N)$ over $y_1$. Plugging this and (94) in (93) readily gives the thesis. $\qquad\square$

**Lemma D.6.** *We have*

$$\left|\varphi(z_1^m)^\top P_{\Phi_{-1}}^\perp \varphi(z_1) - \left(k\left(\sum_{l=2}^{+\infty} \mu_l^2 \alpha^l\right) + \mu_1^2 \left\|P_{\Phi_{-1}}^\perp V_y y_1\right\|_2^2\right)\right| = o(k), \tag{95}$$

*with probability at least $1 - \exp(-c\log^2 N)$ over $V$ and $Z$, where $c$ is an absolute constant.*

*Proof.* An application of Lemma C.3 and Assumption 5.2 gives

$$\|\varphi(z_1)\|_2 = \mathcal{O}\left(\sqrt{k}\right), \qquad \|\varphi(z_1^m)\|_2 = \mathcal{O}\left(\sqrt{k}\right),$$
$$\|Vz_1\|_2 = \mathcal{O}\left(\sqrt{k}\right), \qquad \|Vz_1^m\|_2 = \mathcal{O}\left(\sqrt{k}\right), \tag{96}$$

with probability at least $1 - \exp(-c_1 \log^2 N)$ over $V$, where $c_1$ is an absolute constant. We condition on such high probability event until the end of the proof.

Let's suppose $\mu_0 \neq 0$. Then, we have

$$\left| \varphi(z_1^m)^\top P_{\Phi_{-1}}^\perp \varphi(z_1) - \tilde{\phi}(Vz_1^m)^\top P_{\Phi_{-1}}^\perp \tilde{\phi}(Vz_1) \right| = o(k), \tag{97}$$

with probability at least $1 - \exp(c_2 \log^2 N)$ over $V$ and $Z$, because of (96) and Lemma D.3. Note that (97) trivially holds even when $\mu_0 = 0$, as $\phi \equiv \tilde{\phi}$. Thus, (97) is true in any case with probability at least $1 - \exp(c_2 \log^2 N)$ over $V$ and $Z$.

Furthermore, because of (96) and Lemma D.4, we have

$$\left| \tilde{\phi}(Vz_1^m)^\top P_{\Phi_{-1}} \tilde{\phi}(Vz_1) - \mu_1^2 (Vz_1^m)^\top P_{\Phi_{-1}} (Vz_1) \right| = o(k), \tag{98}$$

with probability at least $1 - \exp(-c_3 \log^2 N)$ over $V$ and $Z$.

Thus, putting (97) and (98) together, and using Lemma D.5, we get

$$\left| \varphi(z_1^m)^\top P_{\Phi_{-1}}^\perp \varphi(z_1) - \left( \tilde{\phi}(Vz_1^m)^\top \tilde{\phi}(Vz_1) - \mu_1^2 (Vz_1^m)^\top (Vz_1) + \mu_1^2 \left\| P_{\Phi_{-1}}^\perp V_y y_1 \right\|_2^2 \right) \right|$$
$$\leq \left| \varphi(z_1^m)^\top P_{\Phi_{-1}}^\perp \varphi(z_1) - \tilde{\phi}(Vz_1^m)^\top P_{\Phi_{-1}}^\perp \tilde{\phi}(Vz_1) \right|$$
$$+ \left| -\tilde{\phi}(Vz_1^m)^\top P_{\Phi_{-1}} \tilde{\phi}(Vz_1) + \mu_1^2 (Vz_1^m)^\top P_{\Phi_{-1}} (Vz_1) \right| \tag{99}$$
$$+ \left| \mu_1^2 (Vz_1^m)^\top P_{\Phi_{-1}}^\perp (Vz_1) - \mu_1^2 \left\| P_{\Phi_{-1}}^\perp V_y y_1 \right\|_2^2 \right| = o(k),$$

with probability at least $1 - \exp(-c_4 \log^2 N)$ over $V$ and $X$ and $x$. To conclude we apply Lemma C.4 setting $t = N$, together with Assumption 5.2, to get

$$\left| \tilde{\phi}(Vz_1^m)^\top \tilde{\phi}(Vz_1) - k \left( \sum_{l=1}^{+\infty} \mu_l^2 \alpha^l \right) \right| = \mathcal{O}\left( \sqrt{k} \left( \sqrt{\frac{k}{d}} + 1 \right) \log N \right) = o(k), \tag{100}$$

and

$$\left| \mu_1^2 (Vz_1^m)^\top (Vz_1) - k\mu_1^2 \alpha \right| = \mathcal{O}\left( \sqrt{k} \left( \sqrt{\frac{k}{d}} + 1 \right) \log N \right) = o(k), \tag{101}$$

which jointly hold with probability at least $1 - \exp(-c_5 \log^2 N)$ over $V$ and $x$.

Applying the triangle inequality to (99), (100), and (101), we get the thesis. $\qquad\square$

**Lemma D.7.** *We have that*

$$\left| \left\| P_{\Phi_{-1}}^\perp \varphi(z_1) \right\|_2^2 - \mathbb{E}_{z_1} \left[ \left\| P_{\Phi_{-1}}^\perp \varphi(z_1) \right\|_2^2 \right] \right| = o(k), \tag{102}$$

$$\left| \varphi(z_1^m)^\top P_{\Phi_{-1}}^\perp \varphi(z_1) - \mathbb{E}_{z_1, z_1^m} \left[ \varphi(z_1^m)^\top P_{\Phi_{-1}}^\perp \varphi(z_1) \right] \right| = o(k), \tag{103}$$

*jointly hold with probability at least $1 - \exp(-c \log^2 N)$ over $z_1$, $V$ and $x$, where $c$ is an absolute constant.*

*Proof.* Let's condition until the end of the proof on both $\|V_x\|_{\mathrm{op}}$ and $\|V_y\|_{\mathrm{op}}$ to be $\mathcal{O}\left(\sqrt{k/d}+1\right)$, which happens with probability at least $1 - e^{-c_1 d}$ by Theorem 4.4.5 of Vershynin (2018). This also implies that $\|V\|_{\mathrm{op}} = \mathcal{O}\left(\sqrt{k/d}+1\right)$.

We indicate with $\nu := \mathbb{E}_{z_1}[\varphi(z_1)] = \mathbb{E}_{z_1^m}[\varphi(z_1^m)] \in \mathbb{R}^k$, and with $\hat{\varphi}(z) := \varphi(z) - \nu$. Note that, as $\varphi$ is a $C\left(\sqrt{k/d}+1\right)$-Lipschitz function, for some constant $C$, and as $z_1$ is Lipschitz concentrated, by Assumption5.2, we have

$$\left|\|\varphi(z_1)\|_2 - \mathbb{E}_{z_1}[\|\varphi(z_1)\|_2]\right| = o\left(\sqrt{k}\right), \tag{104}$$

with probability at least $1 - \exp(-c_2 \log^2 N)$ over $z_1$ and $V$. In addition, by the last statement of Lemma C.3 and Assumption 5.2, we have that $\|\varphi(z_1)\|_2 = \mathcal{O}\left(\sqrt{k}\right)$ with probability $1 - \exp(-c_3 \log^2 N)$ over $V$. Thus, taking the intersection between these two events, we have

$$\mathbb{E}_{z_1}[\|\varphi(z_1)\|_2] = \mathcal{O}\left(\sqrt{k}\right), \tag{105}$$

with probability at least $1 - \exp(-c_4 \log^2 N)$ over $z_1$ and $V$. As this statement is independent of $z_1$, it holds with the same probability just over the probability space of $V$. Then, by Jensen inequality, we have

$$\|\nu\|_2 = \|\mathbb{E}_{z_1}[\varphi(z_1)]\|_2 \leq \mathbb{E}_{z_1}[\|\varphi(z_1)\|_2] = \mathcal{O}\left(\sqrt{k}\right). \tag{106}$$

We can now rewrite the LHS of the first statement as

$$\begin{aligned}
&\left|\left\|P_{\Phi_{-1}}^{\perp}\varphi(z_1)\right\|_2^2 - \mathbb{E}_{z_1}\left[\left\|P_{\Phi_{-1}}^{\perp}\varphi(z_1)\right\|_2^2\right]\right| \\
&= \left|\left\|P_{\Phi_{-1}}^{\perp}(\hat{\varphi}(z_1) + \nu)\right\|_2^2 - \mathbb{E}_{z_1}\left[\left\|P_{\Phi_{-1}}^{\perp}(\hat{\varphi}(z_1) + \nu)\right\|_2^2\right]\right| \\
&= \left|\hat{\varphi}(z_1)^\top P_{\Phi_{-1}}^{\perp}\hat{\varphi}(z_1) + 2\nu^\top P_{\Phi_{-1}}^{\perp}\hat{\varphi}(z_1) - \mathbb{E}_{z_1}\left[\hat{\varphi}(z_1)^\top P_{\Phi_{-1}}^{\perp}\hat{\varphi}(z_1)\right]\right| \\
&\leq \left|\hat{\varphi}(z_1)^\top P_{\Phi_{-1}}^{\perp}\hat{\varphi}(z_1) - \mathbb{E}_{z_1}\left[\hat{\varphi}(z_1)^\top P_{\Phi_{-1}}^{\perp}\hat{\varphi}(z_1)\right]\right| + 2\left|\nu^\top P_{\Phi_{-1}}^{\perp}\hat{\varphi}(z_1)\right|.
\end{aligned} \tag{107}$$

The second term is the inner product between $\hat{\varphi}(z_1)$, a mean-0 sub-Gaussian vector (in the probability space of $z_1$) such that $\|\hat{\varphi}(z_1)\|_{\psi_2} = \mathcal{O}\left(\sqrt{k/d}+1\right)$, and the independent vector $P_{\Phi_{-1}}^{\perp}\nu$, such that $\left\|P_{\Phi_{-1}}^{\perp}\nu\right\|_2 \leq \|\nu\|_2 = \mathcal{O}\left(\sqrt{k}\right)$, because of (106). Thus, by Assumption5.2, we have that

$$\left|\nu^\top P_{\Phi_{-1}}^{\perp}\hat{\varphi}(z_1)\right| = o(k), \tag{108}$$

with probability at least $1 - \exp(-c_5 \log^2 N)$ over $z_1$ and $V$. Then, as $\left(\sqrt{k/d}+1\right)^{-1}\hat{\varphi}(z_1)$ is a mean-0, Lipschitz concentrated random vector (in the probability space of $z_1$), by the general version of the Hanson-Wright inequality given by Theorem 2.3 in Adamczak (2015), we can write

$$\begin{aligned}
&\mathbb{P}\left(\left|\left\|P_{\Phi_{-1}}^{\perp}\hat{\varphi}(z_1)\right\|_2^2 - \mathbb{E}_{z_1}\left[\left\|P_{\Phi_{-1}}^{\perp}\hat{\varphi}(z_1)\right\|_2^2\right]\right| \geq k/\log N\right) \\
&\leq 2\exp\left(-c_6 \min\left(\frac{k^2}{\log^2 N\left((k/d)^2+1\right)\left\|P_{\Phi_{-1}}^{\perp}\right\|_F^2}, \frac{k}{\log N\left(k/d+1\right)\left\|P_{\Phi_{-1}}^{\perp}\right\|_{\mathrm{op}}}\right)\right) \\
&\leq 2\exp\left(-c_6 \min\left(\frac{k}{\log^2 N\left((k/d)^2+1\right)}, \frac{k}{\log N\left(k/d+1\right)}\right)\right) \\
&\leq \exp\left(-c_7 \log^2 N\right),
\end{aligned} \tag{109}$$

where the last inequality comes from Assumption 5.2.This, together with (107) and (108), proves the first part of the statement.

For the second part of the statement, we have

$$
\left| \varphi(z_1^m)^\top P_{\Phi_{-1}}^\perp \varphi(z_1) - \mathbb{E}_{z_1, z_1^m} \left[ \varphi(z_1^m)^\top P_{\Phi_{-1}}^\perp \varphi(z_1) \right] \right|
$$
$$
\leq \left| \hat{\varphi}(z_1^m)^\top P_{\Phi_{-1}}^\perp \hat{\varphi}(z_1) - \mathbb{E}_{z_1, z_1^m} \left[ \hat{\varphi}(z_1^m)^\top P_{\Phi_{-1}}^\perp \hat{\varphi}(z_1) \right] \right| + \left| \nu^\top P_{\Phi_{-1}}^\perp \hat{\varphi}(z_1) \right| + \left| \nu^\top P_{\Phi_{-1}}^\perp \hat{\varphi}(z_1^m) \right|.
$$
(110)

Following the same argument that led to (108), we obtain

$$
\left| \nu^\top P_{\Phi_{-1}}^\perp \hat{\varphi}(z_1^m) \right| = o(k),
$$
(111)

with probability at least $1 - \exp(-c_8 \log^2 N)$ over $z_1^m$ and $V$. Let us set

$$
P_2 := \frac{1}{2} \left( \begin{array}{c|c} 0 & P_{\Phi_{-1}}^\perp \\ \hline P_{\Phi_{-1}}^\perp & 0 \end{array} \right), \qquad V_2 := \left( \begin{array}{c|c|c} V_x & V_y & 0 \\ \hline 0 & V_y & V_x \end{array} \right),
$$
(112)

and

$$
\hat{\varphi}_2 := \phi \left( V_2 [x_1, y_1, x]^\top \right) - \mathbb{E}_{x_1, y_1, x} \left[ \phi \left( V_2 [x_1, y_1, x]^\top \right) \right] \equiv [\hat{\varphi}(z_1), \hat{\varphi}(z_1^m)]^\top.
$$
(113)

We have that $\|P_2\|_{\mathrm{op}} \leq 1$, $\|P_2\|_F^2 \leq k$, $\|V_2\|_{\mathrm{op}} \leq 2 \|V_x\|_{\mathrm{op}} + 2 \|V_y\|_{\mathrm{op}} = \mathcal{O}\left( \sqrt{k/d} + 1 \right)$, and that $[x_1, y_1, x]^\top$ is a Lipschitz concentrated random vector in the joint probability space of $z_1$ and $z_1^m$, which follows from applying Lemma C.2 twice. Also, we have

$$
\hat{\varphi}(z_1^m)^\top P_{\Phi_{-1}}^\perp \hat{\varphi}(z_1) = \hat{\varphi}_2^\top P_2 \hat{\varphi}_2.
$$
(114)

Thus, as $\left( \sqrt{k/d} + 1 \right)^{-1} \hat{\varphi}_2$ is a mean-0, Lipschitz concentrated random vector (in the probability space of $z_1$ and $z_1^m$), again by the general version of the Hanson-Wright inequality given by Theorem 2.3 in Adamczak (2015), we can write

$$
\mathbb{P}\left( \left| \hat{\varphi}_2^\top P_2 \hat{\varphi}_2 - \mathbb{E}_{z_1, z_1^m} \left[ \hat{\varphi}_2^\top P_2 \hat{\varphi}_2 \right] \right| \geq k / \log N \right)
$$
$$
\leq 2 \exp \left( -c_9 \min \left( \frac{k^2}{\log^2 N \left( (k/d)^2 + 1 \right) \|P_2\|_F^2}, \frac{k}{\log N \left( k/d + 1 \right) \|P_2\|_{\mathrm{op}}} \right) \right)
$$
$$
\leq 2 \exp \left( -c_9 \min \left( \frac{k}{\log^2 N \left( (k/d)^2 + 1 \right)}, \frac{k}{\log N \left( k/d + 1 \right)} \right) \right)
$$
$$
\leq \exp \left( -c_{10} \log^2 N \right),
$$
(115)

where the last inequality comes from Assumption 5.2. This, together with (110), (108), (111), and (114), proves the second part of the statement, and therefore the desired result.

$\square$

Finally, we are ready to give the proof of Theorem 5.4.

*Proof of Theorem 5.4.* We will prove the statement for the following definition of $\gamma_{\mathrm{RF}}$, independent from $z_1$ and $z_1^m$,

$$
\gamma_{\mathrm{RF}} := \frac{\mathbb{E}_{z_1, z_1^m} \left[ \varphi(z_1^m)^\top P_{\Phi_{-1}}^\perp \varphi(z_1) \right]}{\mathbb{E}_{z_1} \left[ \left\| P_{\Phi_{-1}}^\perp \varphi(z_1) \right\|_2^2 \right]}.
$$
(116)

By Lemma B.1 and D.2, we have

$$
\left\| P_{\Phi_{-1}}^\perp \varphi(z) \right\|_2^2 = \Omega(k)
$$
(117)

with probability at least $1 - \exp(-c_1 \log^2 N)$ over $V$, $Z_{-1}$ and $z$. This, together with Lemma D.7, gives

$$
\left| \frac{\varphi(z_1^m)^\top P_{\Phi_{-1}}^\perp \varphi(z_1)}{\left\| P_{\Phi_{-1}}^\perp \varphi(z_1) \right\|_2^2} - \frac{\mathbb{E}_{z_1, z_1^m} \left[ \varphi(z_1^m)^\top P_{\Phi_{-1}}^\perp \varphi(z_1) \right]}{\mathbb{E}_{z_1} \left[ \left\| P_{\Phi_{-1}}^\perp \varphi(z_1) \right\|_2^2 \right]} \right| = o(1),
$$
(118)

with probability at least $1 - \exp(-c_2 \log^2 N)$ over $V$, $Z$ and $x$, which proves the first part of the statement.

The upper-bound on $\gamma_{\mathrm{RF}}$ can be obtained applying Cauchy-Schwarz twice

$$
\begin{aligned}
\frac{\mathbb{E}_{z_1, z_1^m} \left[ \varphi(z_1^m)^\top P_{\Phi_{-1}}^\perp \varphi(z_1) \right]}{\mathbb{E}_{z_1} \left[ \left\| P_{\Phi_{-1}}^\perp \varphi(z_1) \right\|_2^2 \right]} &\leq \frac{\mathbb{E}_{z_1, z_1^m} \left[ \left\| P_{\Phi_{-1}}^\perp \varphi(z_1^m) \right\|_2 \left\| P_{\Phi_{-1}}^\perp \varphi(z_1) \right\|_2 \right]}{\mathbb{E}_{z_1} \left[ \left\| P_{\Phi_{-1}}^\perp \varphi(z_1) \right\|_2^2 \right]} \\
&\leq \frac{\sqrt{\mathbb{E}_{z_1^m} \left[ \left\| P_{\Phi_{-1}}^\perp \varphi(z_1^m) \right\|_2^2 \right]} \sqrt{\mathbb{E}_{z_1} \left[ \left\| P_{\Phi_{-1}}^\perp \varphi(z_1) \right\|_2^2 \right]}}{\mathbb{E}_{z_1} \left[ \left\| P_{\Phi_{-1}}^\perp \varphi(z_1) \right\|_2^2 \right]} = 1.
\end{aligned}
\tag{119}
$$

Let's now focus on the lower bound. By Assumption 5.2 and Lemma C.4 (in which we consider the degenerate case $\alpha = 1$ and set $t = N$), we have

$$
\left| \left\| \tilde{\phi}(V z_1) \right\|_2^2 - k \sum_{l=1}^{+\infty} \mu_l^2 \right| = o(k),
\tag{120}
$$

with probability at least $1 - \exp(-c_3 \log^2 N)$ over $V$ and $z_1$. Then, a few applications of the triangle inequality give

$$
\begin{aligned}
\frac{\mathbb{E}_{z_1, z_1^m} \left[ \varphi(z_1^m)^\top P_{\Phi_{-1}}^\perp \varphi(z_1) \right]}{\mathbb{E}_{z_1} \left[ \left\| P_{\Phi_{-1}}^\perp \varphi(z_1) \right\|_2^2 \right]} &\geq \frac{\varphi(z_1^m)^\top P_{\Phi_{-1}}^\perp \varphi(z_1)}{\left\| P_{\Phi_{-1}}^\perp \varphi(z_1) \right\|_2^2} - o(1) \\
&\geq \frac{\varphi(z_1^m)^\top P_{\Phi_{-1}}^\perp \varphi(z_1)}{\left\| P_{\Phi_{-1}}^\perp \tilde{\varphi}(z_1) \right\|_2^2} - o(1) \\
&\geq \frac{k \left( \sum_{l=2}^{+\infty} \mu_l^2 \alpha^l \right) + \mu_1^2 \left\| P_{\Phi_{-1}}^\perp V_y y_1 \right\|_2^2}{\| \tilde{\varphi}(z_1) \|_2^2} - o(1) \\
&\geq \frac{k \left( \sum_{l=2}^{+\infty} \mu_l^2 \alpha^l \right) + \mu_1^2 \left\| P_{\Phi_{-1}}^\perp V_y y_1 \right\|_2^2}{k \sum_{l=1}^{+\infty} \mu_l^2} - o(1) \\
&\geq \frac{\sum_{l=2}^{+\infty} \mu_l^2 \alpha^l}{\sum_{l=1}^{+\infty} \mu_l^2} - o(1),
\end{aligned}
\tag{121}
$$

where the first inequality is a consequence of (118), the second of Lemma D.3 and (117), the third of Lemma D.6 and again (117), and the fourth of (120), and they jointly hold with probability $1 - \exp(-c_4 \log^2 N)$ over $V$, $Z_{-1}$ and $z_1$. Again, as the statement does not depend on $z_1$, we can conclude that it holds with the same probability only over the probability spaces of $V$ and $Z_{-1}$, and the thesis readily follows. $\qquad\square$

## E  PROOFS FOR NTK REGRESSION

In this section, we will indicate with $Z \in \mathbb{R}^{N \times d}$ the data matrix, such that its rows are sampled independently from $\mathcal{P}_Z$ (see Assumption 5.1). We denote by $W \in \mathbb{R}^{k \times d}$ the weight matrix at initialization, such that $W_{ij} \sim_{\mathrm{i.i.d.}} \mathcal{N}(0, 1/d)$. Thus, the feature map is given by (see (23))

$$
\varphi(z) := z \otimes \phi'(Wz) \in \mathbb{R}^{dk},
\tag{122}
$$

where $\phi'$ is the derivative of the activation function $\phi$, applied component-wise to the vector $Wz$. We use the shorthands $\Phi := Z * \phi'(ZW^\top) \in \mathbb{R}^{N \times p}$ and $K := \Phi\Phi^\top \in \mathbb{R}^{N \times N}$, where $*$ denotes the Khatri-Rao product, defined in Appendix A. We indicate with $\Phi_{-1} \in \mathbb{R}^{(N-1) \times k}$ the matrix

$\Phi$ without the first row, and we define $K_{-1} := \Phi_{-1}\Phi_{-1}^{\top}$. We call $P_\Phi$ the projector over the span of the rows of $\Phi$, and $P_{\Phi_{-1}}$ the projector over the span of the rows of $\Phi_{-1}$. We use the notations $\tilde{\varphi}(z) := \varphi(z) - \mathbb{E}_W[\varphi(z)]$ and $\tilde{\Phi}_{-1} := \Phi_{-1} - \mathbb{E}_W[\Phi_{-1}]$ to indicate the centered feature map and matrix respectively, where the centering is with respect to $W$. We indicate with $\mu'_l$ the $l$-th Hermite coefficient of $\phi'$. We use the notation $z_1^m = [x, y_1]$, where $x \sim \mathcal{P}_X$ is sampled independently from $V$ and $Z$. We define $\alpha = d_y/d$. Throughout this section, for compactness, we drop the subscripts "NTK" from these quantities, as we will only treat the proofs related to Section 6. Again for the sake of compactness, we will not re-introduce such quantities in the statements or the proofs of the following lemmas.

The content of this section can be summarized as follows:

- We start by presenting a sketch of the proof.

- In Lemma E.1, we prove the lower bound on the smallest eigenvalue of $K$, adapting to our settings the main result of Bombari et al. (2022b).

- In Lemma E.5, we treat separately a term that derives from $\mathbb{E}_W[\phi'(Wz)] = \mu'_0 \mathbf{1}_k$, showing that we can *center* the derivative of the activation function (Lemma E.9), without changing our final statement in Theorem 6.3. This step is necessary only if $\mu'_0 \neq 0$. Our proof tackles the problem proving the thesis on a set of "perturbed" inputs $\bar{Z}_{-1}(\delta)$ (Lemma E.4), critically exploiting the non degenerate behaviour of their rows (Lemma E.3), and transfers the result on the original term, using continuity arguments with respect to the perturbation (Lemma E.2).

- In Lemma E.8, we show that the centered features $\tilde{\varphi}(z_1)$ and $\tilde{\varphi}(z_1^m)$ have a negligible component in the space spanned by the rows of $\Phi_{-1}$. To achieve this, we exploit the bound proved in Lemma E.7.

- To conclude, we prove Theorem 6.3, exploiting also the concentration result provided in Lemma E.6.

**Proof sketch.** The argument is more direct than for the RF model since, in this case, we are able to express $\gamma_{\mathrm{NTK}}$ in closed form. The *first step* is to *center the feature map* $\varphi$, which gives

$$\frac{\varphi(z_1^m)^{\top} P_{\Phi_{-1}}^{\perp} \varphi(z_1)}{\left\| P_{\Phi_{-1}}^{\perp} \varphi(z_1) \right\|_2^2} \simeq \frac{\tilde{\varphi}(z_1^m)^{\top} P_{\Phi_{-1}}^{\perp} \tilde{\varphi}(z_1)}{\left\| P_{\Phi_{-1}}^{\perp} \tilde{\varphi}(z_1) \right\|_2^2}. \tag{123}$$

While a similar step appeared in the analysis of the RF model, its implementation for NTK requires a different strategy. In particular, we exploit that the samples $z_1$ and $z_1^m$ are *approximately* contained in the span of the rows of $Z_{-1}$ (see Lemma E.4). As the rows of $Z_{-1}$ may not *exactly* span all $\mathbb{R}^d$, we resort to an *approximation* by adding a small amount of independent noise to every entry of $Z_{-1}$. The resulting perturbed dataset $\bar{Z}_{-1}$ satisfies $\mathrm{Span}\{\mathrm{rows}(\bar{Z}_{-1})\} = \mathbb{R}^d$ (see Lemma E.3), and we conclude via a continuity argument with respect to the magnitude of the perturbation (see Lemmas E.2 and E.5).

The *second step* is to upper bound the terms $\left| \tilde{\varphi}(z_1^m)^{\top} P_{\Phi_{-1}} \tilde{\varphi}(z_1) \right|$ and $\left\| P_{\Phi_{-1}} \tilde{\varphi}(z_1) \right\|_2^2$, showing they have *negligible magnitude*, which gives

$$\frac{\tilde{\varphi}(z_1^m)^{\top} P_{\Phi_{-1}}^{\perp} \tilde{\varphi}(z_1)}{\left\| P_{\Phi_{-1}}^{\perp} \tilde{\varphi}(z_1) \right\|_2^2} \simeq \frac{\tilde{\varphi}(z_1^m)^{\top} \tilde{\varphi}(z_1)}{\|\tilde{\varphi}(z_1)\|_2^2}. \tag{124}$$

This is a consequence of the fact that, if $z \sim \mathcal{P}_Z$ is independent from $Z_{-1}$, then $\tilde{\varphi}(z)$ is roughly orthogonal to $\mathrm{Span}\{\mathrm{rows}(\Phi_{-1})\}$, see Lemma E.8.

Finally, the *third step* is to show the *concentration* over $W_0$ of the numerator and denominator of the RHS of (124), see Lemma E.6. This allows us to conclude that

$$\frac{\tilde{\varphi}(z_1^m)^{\top} \tilde{\varphi}(z_1)}{\|\tilde{\varphi}(z_1)\|_2^2} \simeq \alpha \frac{\sum_{l=1}^{+\infty} \mu_l'^2 \alpha^i}{\sum_{l=1}^{+\infty} \mu_l'^2} > 0. \tag{125}$$

The RHS of (125) is strictly positive as $\alpha > 0$ and $\phi$ is non-linear by Assumption 6.2.

**Lemma E.1.** *We have that*

$$\lambda_{\min}\left(K\right) = \Omega(kd), \tag{126}$$

*with probability at least $1 - Ne^{-c\log^2 k} - e^{-c\log^2 N}$ over $Z$ and $W$, where $c$ is an absolute constant.*

*Proof.* The result follows from Theorem 3.1 of Bombari et al. (2022b). Notice that our assumptions on the data distribution $\mathcal{P}_Z$ are stronger, and that our initialization of the very last layer (which differs from the Gaussian initialization in Bombari et al. (2022b)) does not change the result. Assumption 6.1, *i.e.*, $k = \mathcal{O}\left(d\right)$, satisfies the *loose pyramidal topology* condition (cf. Assumption 2.4 in Bombari et al. (2022b)), and Assumption 6.1 is the same as Assumption 2.5 in Bombari et al. (2022b). An important difference is that we do not assume the activation function $\phi$ to be Lipschitz anymore. This, however, stops being a necessary assumption since we are working with a 2-layer neural network, and $\phi$ doesn't appear in the expression of NTK. $\square$

**Lemma E.2.** *Let $A \in \mathbb{R}^{(N-1)\times d}$ be a generic matrix, and let $\bar{Z}_{-1}(\delta)$ and $\bar{\Phi}_{-1}(\delta)$ be defined as*

$$\bar{Z}_{-1}(\delta) := Z_{-1} + \delta A, \tag{127}$$

$$\bar{\Phi}_{-1}(\delta) := \bar{Z}_{-1}(\delta) * \phi'\left(Z_{-1}W^\top\right). \tag{128}$$

*Let $\bar{P}_{\Phi_{-1}}(\delta) \in \mathbb{R}^{dk\times dk}$ be the projector over the Span of the rows of $\bar{\Phi}_{-1}(\delta)$. Then, we have that $\bar{P}_{\Phi_{-1}}^\perp(\delta)$ is continuous in $\delta = 0$ with probability at least $1 - Ne^{-c\log^2 k} - e^{-c\log^2 N}$ over $Z$ and $W$, where $c$ is an absolute constant and where the continuity is with respect to $\|\cdot\|_{\mathrm{op}}$.*

*Proof.* In this proof, when we say that a matrix is continuous with respect to $\delta$, we always intend with respect to the operator norm $\|\cdot\|_{\mathrm{op}}$. Then, $\bar{\Phi}_{-1}(\delta)$ is continuous in $0$, as

$$\left\|\bar{\Phi}_{-1}(\delta) - \bar{\Phi}_{-1}(0)\right\|_{\mathrm{op}} = \left\|\delta A * \phi'\left(Z_{-1}W^\top\right)\right\|_{\mathrm{op}} \le \delta \left\|A\right\|_{\mathrm{op}} \max_{2\le i\le N} \left\|\phi'\left(Wz_i\right)\right\|_2, \tag{129}$$

where the second step follows from Equation (3.7.13) in Johnson (1990).

By Weyl's inequality, this also implies that $\lambda_{\min}\left(\bar{\Phi}_{-1}(\delta)\bar{\Phi}_{-1}(\delta)^\top\right)$ is continuous in $\delta = 0$. Recall that, by Lemma E.1, $\det\left(\bar{\Phi}_{-1}(0)\bar{\Phi}_{-1}(0)^\top\right) \equiv \det\left(\Phi_{-1}\Phi_{-1}^\top\right) \ne 0$ with probability at least $1 - Ne^{-c\log^2 k} - e^{-c\log^2 N}$ over $Z$ and $W$. This implies that $\left(\bar{\Phi}_{-1}(\delta)\bar{\Phi}_{-1}(\delta)^\top\right)^{-1}$ is also continuous, as for every invertible matrix $M$ we have $M^{-1} = \mathrm{Adj}(M)/\det(M)$ (where $\mathrm{Adj}(M)$ denotes the Adjugate of the matrix $M$), and both $\mathrm{Adj}(\cdot)$ and $\det(\cdot)$ are continuous mappings. Thus, as $\bar{P}_{\Phi_{-1}}(0) = \bar{\Phi}_{-1}(0)^\top \left(\bar{\Phi}_{-1}(0)\bar{\Phi}_{-1}(0)^\top\right)^{-1} \bar{\Phi}_{-1}(0)$ (see (33)), we also have the continuity of $\bar{P}_{\Phi_{-1}}(\delta)$ in $\delta = 0$, which gives the thesis. $\square$

**Lemma E.3.** *Let $A \in \mathbb{R}^{(N-1)\times d}$ be a matrix with entries sampled independently (between each other and from everything else) from a standard Gaussian distribution. Then, for every $\delta > 0$, with probability 1 over $A$, the rows of $\bar{Z}_{-1} := Z_{-1} + \delta A$ span $\mathbb{R}^d$.*

*Proof.* As $N - 1 \ge d$, by Assumption 6.1, negating the thesis would imply that the rows of $\bar{Z}_{-1}$ are linearly dependent, and that they belong to a subspace with dimension at most $d - 1$. This would imply that there exists a row of $\bar{Z}_{-1}$, call it $\bar{z}_j$, such that $\bar{z}_j$ belongs to the space spanned by all the other rows of $\bar{Z}_{-1}$, with dimension at most $d - 1$. This means that $A_{j:}$ has to belong to an affine space with the same dimension, which we can consider fixed, as it's not a function of the random vector $A_{j:}$, but only of $Z_{-1}$ and $\{A_{i:}\}_{i\ne j}$. As the entries of $A_{j:}$ are sampled independently from a standard Gaussian distribution, this happens with probability 0. $\square$

**Lemma E.4.** *Let $A \in \mathbb{R}^{(N-1)\times d}$ be a matrix with entries sampled independently (between each other and from everything else) from a standard Gaussian distribution. Let $\bar{Z}_{-1}(\delta) := Z_{-1} + \delta A$ and $\bar{\Phi}_{-1}(\delta) := \bar{Z}_{-1}(\delta) * \phi'\left(Z_{-1}W^\top\right)$. Let $\bar{P}_{\Phi_{-1}}(\delta) \in \mathbb{R}^{dk\times dk}$ be the projector over the Span of the rows of $\bar{\Phi}_{-1}(\delta)$. Let $\mu_0' \ne 0$. Then, for $z \sim \mathcal{P}_Z$, and for any $\delta > 0$, we have,*

$$\left\|\bar{P}_{\Phi_{-1}}^\perp(\delta)\left(z \otimes \mathbf{1}_k\right)\right\|_2 = o(\sqrt{dk}), \tag{130}$$

*with probability at least $1 - \exp(-c\log^2 N)$ over $Z$, $W$, and $A$, where $c$ is an absolute constant.*

*Proof.* Let $B_{-1} := \phi'(Z_{-1}W^\top) \in \mathbb{R}^{(N-1)\times k}$. Notice that, for any $\zeta \in \mathbb{R}^{N-1}$, the following identity holds

$$\bar{\Phi}_{-1}^\top(\delta)\zeta = \left(\bar{Z}_{-1}(\delta) * B_{-1}\right)^\top \zeta = \left(\bar{Z}_{-1}^\top(\delta)\zeta\right) \otimes \left(B_{-1}^\top \mathbf{1}_{N-1}\right). \tag{131}$$

Note that $B_{-1}^\top = \mu_0' \mathbf{1}_k \mathbf{1}_{N-1}^\top + \tilde{B}_{-1}^\top$, where $\tilde{B}_{-1}^\top = \phi'(WZ_{-1}^\top) - \mathbb{E}_W\left[\phi'(WZ_{-1}^\top)\right]$ is a $k \times (N-1)$ matrix with i.i.d. and mean-0 rows. For an argument equivalent to the one used for (79) and (80), we have

$$\left\|\tilde{B}_{-1}^\top\right\|_{\mathrm{op}} = \mathcal{O}\left(\left(\sqrt{k} + \sqrt{N}\right)\left(\sqrt{N/d} + 1\right)\right), \tag{132}$$

with probability at least $1 - \exp(-c\log^2 N)$ over $Z_{-1}$ and $W$. Thus, we can write

$$B_{-1}^\top \frac{\mathbf{1}_{N-1}}{\mu_0'(N-1)} = \left(\mu_0' \mathbf{1}_k \mathbf{1}_{N-1}^\top + \tilde{B}_{-1}^\top\right)\frac{\mathbf{1}_{N-1}}{\mu_0'(N-1)} = \mathbf{1}_k + \tilde{B}_{-1}^\top \frac{\mathbf{1}_{N-1}}{\mu_0'(N-1)} =: \mathbf{1}_k + v, \tag{133}$$

where we have

$$\|v\|_2 \le \left\|\tilde{B}_{-1}^\top\right\|_{\mathrm{op}} \left\|\frac{\mathbf{1}_{N-1}}{\mu_0'(N-1)}\right\|_2 = \mathcal{O}\left(\left(\sqrt{k/N} + 1\right)\left(\sqrt{N/d} + 1\right)\right) = o(\sqrt{k}), \tag{134}$$

where the last step is a consequence of Assumption 6.1. Plugging (133) in (131) we get

$$\frac{1}{\mu_0'(N-1)}\bar{\Phi}_{-1}^\top(\delta)\zeta = \frac{1}{\mu_0'(N-1)}\left(\bar{Z}_{-1}(\delta) * B_{-1}\right)^\top \zeta = \left(\bar{Z}_{-1}^\top(\delta)\zeta\right) \otimes \left(\mathbf{1}_k + v\right). \tag{135}$$

By Lemma E.3, we have that the rows of $\bar{Z}_{-1}(\delta)$ span $\mathbb{R}^d$, with probability 1 over $A$. Thus, conditioning on this event, we can set $\zeta$ to be a vector such that $z = \bar{Z}_{-1}^\top(\delta)\zeta$. We can therefore rewrite the previous equation as

$$\frac{1}{\mu_0'(N-1)}\bar{\Phi}_{-1}^\top(\delta)\zeta = z \otimes \mathbf{1}_k + z \otimes v. \tag{136}$$

Thus, we can conclude

$$\begin{aligned}
\left\|\bar{P}_{\Phi_{-1}}^\perp(\delta)(z \otimes \mathbf{1}_k)\right\|_2 &= \left\|P_{\Phi_{-1}}^\perp\left(\frac{\Phi_{-1}^\top(\delta)\zeta}{\mu_0'(N-1)} - z \otimes v\right)\right\|_2 \\
&\le \left\|\bar{P}_{\Phi_{-1}}^\perp(\delta)\Phi_{-1}^\top(\delta)\frac{\zeta}{\mu_0'(N-1)}\right\|_2 + \|z \otimes v\|_2 \\
&= \|z\|_2 \|v\|_2 = o(\sqrt{dk}),
\end{aligned} \tag{137}$$

where in the second step we use the triangle inequality, in the third step we use that $\Phi_{-1}^\top(\delta) = \bar{P}_{\Phi_{-1}}(\delta)\Phi_{-1}^\top(\delta)$, and in the last step we use (134). The desired result readily follows. $\square$

**Lemma E.5.** *Let $\mu_0' \ne 0$. Then, for any $z \in \mathbb{R}^d$, we have,*

$$\left\|P_{\Phi_{-1}}^\perp(z \otimes \mathbf{1}_k)\right\|_2 = o(\sqrt{dk}), \tag{138}$$

*with probability at least $1 - Ne^{-c\log^2 k} - e^{-c\log^2 N}$ over $Z$ and $W$, where $c$ is an absolute constant.*

*Proof.* Let $A \in \mathbb{R}^{(N-1)\times d}$ be a matrix with entries sampled independently (between each other and from everything else) from a standard Gaussian distribution. Let $\bar{Z}_{-1}(\delta) := Z_{-1} + \delta A$ and $\bar{\Phi}_{-1}(\delta) := \bar{Z}_{-1}(\delta) * \phi'\left(Z_{-1}W^\top\right)$. Let $\bar{P}_{\Phi_{-1}}(\delta) \in \mathbb{R}^{dk \times dk}$ be the projector over the Span of the rows of $\bar{\Phi}_{-1}(\delta)$.

By triangle inequality, we can write

$$\left\|P_{\Phi_{-1}}^\perp(z \otimes \mathbf{1}_k)\right\|_2 \le \left\|P_{\Phi_{-1}}^\perp - \bar{P}_{\Phi_{-1}}^\perp(\delta)\right\|_{\mathrm{op}} \|z \otimes \mathbf{1}_k\|_2 + \left\|\bar{P}_{\Phi_{-1}}^\perp(\delta)(z \otimes \mathbf{1}_k)\right\|_2. \tag{139}$$

Because of Lemma E.2, with probability at least $1 - Ne^{-c \log^2 k} - e^{-c \log^2 N}$ over $Z$ and $W$, $\bar{P}_{\Phi_{-1}}^{\perp}(\delta)$ is continuous in $\delta = 0$, with respect to $\|\cdot\|_{\text{op}}$. Thus, there exists $\delta^* > 0$ such that, for every $\delta \in [0, \delta^*]$,

$$\left\| P_{\Phi_{-1}}^{\perp} - \bar{P}_{\Phi_{-1}}^{\perp}(\delta) \right\|_{\text{op}} \equiv \left\| \bar{P}_{\Phi_{-1}}^{\perp}(0) - \bar{P}_{\Phi_{-1}}^{\perp}(\delta) \right\|_{\text{op}} < \frac{1}{N}. \tag{140}$$

Hence, setting $\delta = \delta^*$ in (139), we get

$$\begin{aligned}
\left\| P_{\Phi_{-1}}^{\perp} \left( z \otimes \mathbf{1}_k \right) \right\|_2 &\leq \left\| P_{\Phi_{-1}}^{\perp} - \bar{P}_{\Phi_{-1}}^{\perp}(\delta^*) \right\|_{\text{op}} \left\| z \otimes \mathbf{1}_k \right\|_2 + \left\| \bar{P}_{\Phi_{-1}}^{\perp}(\delta^*) \left( z \otimes \mathbf{1}_k \right) \right\|_2 \\
&\leq \|z\|_2 \|\mathbf{1}_k\|_2 / N + \left\| \bar{P}_{\Phi_{-1}}^{\perp}(\delta^*) \left( z \otimes \mathbf{1}_k \right) \right\|_2 \\
&= o(\sqrt{dk}),
\end{aligned} \tag{141}$$

where the last step is a consequence of Lemma E.4, and it holds with probability at least $1 - \exp(-c \log^2 N)$ over $Z$, $W$, and $A$. As the LHS of the previous equation doesn't depend on $A$, the statements holds with the same probability, just over the probability spaces of $Z$ and $W$, which gives the desired result. $\qquad\square$

**Lemma E.6.** *We have*

$$\left| \frac{\tilde{\varphi}(z_1^m)^\top \tilde{\varphi}(z_1)}{\|\tilde{\varphi}(z_1)\|_2^2} - \alpha \frac{\sum_{l=1}^{+\infty} \mu_l'^2 \alpha^i}{\sum_{l=1}^{+\infty} \mu_l'^2} \right| = o(1), \tag{142}$$

*with probability at least $1 - \exp(-c \log^2 N) - \exp(-c \log^2 k)$ over $W$ and $z_1$, where $c$ is an absolute constant. With the same probability, we also have*

$$\tilde{\varphi}(z_1^m)^\top \tilde{\varphi}(z_1) = \Theta(dk), \qquad \|\tilde{\varphi}(z_1)\|_2^2 = \Theta(dk). \tag{143}$$

*Proof.* We have

$$\|\tilde{\varphi}(z_1)\|_2^2 = \left\| z_1 \otimes \tilde{\phi}'(W z_1) \right\|_2^2 = \|z_1\|_2^2 \left\| \tilde{\phi}'(W z_1) \right\|_2^2 = d \left\| \tilde{\phi}'(W z_1) \right\|_2^2. \tag{144}$$

By Assumption 6.1 and Lemma C.4 (in which we consider the degenerate case $\alpha = 1$ and set $t = k$), we have

$$\left| \left\| \tilde{\phi}'(W z_1) \right\|_2^2 - k \sum_{l=1}^{+\infty} \mu_l'^2 \right| = o(k), \tag{145}$$

with probability at least $1 - \exp(-c \log^2 k)$ over $W$ and $z_1$. Thus, we have

$$\left| \|\tilde{\varphi}(z_1)\|_2^2 - dk \sum_{l=1}^{+\infty} \mu_l'^2 \right| = o(dk). \tag{146}$$

Notice that the second term in the modulus is $\Theta(dk)$, since the $\mu_l'$-s cannot be all 0, because of Assumption 6.2; this shows that $\|\tilde{\varphi}(z_1)\|_2^2 = \Theta(dk)$.

Similarly, we can write

$$\tilde{\varphi}(z_1^m)^\top \tilde{\varphi}(z_1) = \left( z_1^\top z_1^m \right) \left( \tilde{\phi}'(W z_1)^\top \tilde{\phi}'(W z_1^m) \right). \tag{147}$$

We have

$$\left| z_1^\top z_1^m - \alpha d \right| = \left| x_1^\top x \right| \leq \sqrt{d_x} \log d = o(d), \tag{148}$$

where the inequality holds with probability at least $1 - \exp(-c_1 \log^2 d) \geq 1 - \exp(-c_2 \log^2 N)$ over $x_1$, as we are taking the inner product of two independent and sub-Gaussian vectors with norm $\sqrt{d_x}$. Furthermore, again by Assumption6.1 and Lemma C.4, we have

$$\left| \tilde{\phi}'(W z_1)^\top \tilde{\phi}'(W z_1^m) - k \sum_{l=1}^{+\infty} \mu_l'^2 \alpha^l \right| = o(k), \tag{149}$$

with probability at least $1 - \exp(-c_3 \log^2 k)$ over $W$ and $z_1$. Notice that the second term in the modulus is $\Theta(k)$, because of Assumption 6.2.

Thus, putting (147), (148) and (149) together, we get

$$\left| \tilde{\varphi}(z_1^m)^\top \tilde{\varphi}(z_1) - dk\alpha \sum_{l=1}^{+\infty} {\mu'_l}^2 \alpha^l \right| = o(dk), \tag{150}$$

with probability at least $1 - \exp(-c_3 \log^2 k) - \exp(-c_2 \log^2 N)$ over $W$ and $z_1$; this shows that $\tilde{\varphi}(z_1^m)^\top \tilde{\varphi}(z_1) = \Theta(dk)$.

Finally, merging (150) with (146) and applying triangle inequality, (142) follows and the proof is complete. $\qquad\square$

**Lemma E.7.** *Let $z \sim \mathcal{P}_Z$ be sampled independently from $Z_{-1}$. Then,*
$$\|\Phi_{-1}\tilde{\varphi}(z)\|_2 = o\,(dk)\,, \tag{151}$$
*with probability at least $1 - \exp(-c \log^2 N)$ over $W$ and $z$, where $c$ is an absolute constant.*

*Proof.* Let's look at the $i$-th entry of the vector $\Phi_{-1}\tilde{\varphi}(z)$, *i.e.*,
$$\varphi(z_{i+1})^\top \tilde{\varphi}(z) = \left(z_{i+1}^\top z\right)\left(\phi'(Wz_{i+1})^\top \tilde{\phi}'(Wz)\right). \tag{152}$$

As $z$ and $z_{i+1}$ are sub-Gaussian and independent with norm $\sqrt{d}$, we can write $\left|z^\top z_{i+1}\right| = \mathcal{O}\left(\sqrt{d}\log N\right)$ with probability at least $1 - \exp(-c \log^2 N)$ over $z$. We will condition on such high probability event until the end of the proof.

By Lemma C.3, setting $t = N$, we have
$$\left| \phi'(Wz_{i+1})^\top \tilde{\phi}'(Wz) - \mathbb{E}_W\left[\phi'(Wz_{i+1})^\top \tilde{\phi}'(Wz)\right]\right| = \mathcal{O}\left(\sqrt{k}\log N\right), \tag{153}$$

with probability at least $1 - \exp(-c_1 \log^2 N)$ over $W$. Exploiting the Hermite expansion of $\phi'$ and $\tilde{\phi}'$, we have

$$\begin{aligned}
\left|\mathbb{E}_W\left[\phi'(Wz_{i+1})^\top \tilde{\phi}'(Wz)\right]\right| &\leq k \sum_{l=1}^{+\infty} {\mu'_l}^2 \left(\frac{\left|z_{i+1}^\top z\right|}{\|z_{i+1}\|_2 \|z\|_2}\right)^l \\
&\leq k \max_l {\mu'_l}^2 \sum_{l=1}^{+\infty} \left(\frac{\left|z_{i+1}^\top z\right|}{\|z_{i+1}\|_2 \|z\|_2}\right)^l \\
&= k \max_l {\mu'_l}^2 \frac{\left|z_{i+1}^\top z\right|}{\|z_{i+1}\|_2 \|z\|_2} \frac{1}{1 - \frac{\left|z_{i+1}^\top z\right|}{\|z_{i+1}\|_2 \|z\|_2}} \\
&\leq 2k \max_l {\mu'_l}^2 \frac{\left|z_{i+1}^\top z\right|}{\|z_{i+1}\|_2 \|z\|_2} = \mathcal{O}\left(\frac{k\log N}{\sqrt{d}}\right).
\end{aligned} \tag{154}$$

Putting together (153) and (154), and applying triangle inequality, we get
$$\left|\phi'(Wz_{i+1})^\top \tilde{\phi}'(Wz)\right| = \mathcal{O}\left(\sqrt{k}\log N + \frac{k\log N}{\sqrt{d}}\right) = \mathcal{O}\left(\sqrt{k}\log N\right), \tag{155}$$

where the last step is a consequence of Assumption 6.1. Comparing this last result with (152), we obtain
$$\left|\varphi(z_{i+1})^\top \tilde{\varphi}(z)\right| = \mathcal{O}\left(\sqrt{dk}\log^2 N\right), \tag{156}$$

with probability at least $1 - \exp(-c_2 \log^2 N)$ over $W$ and $z$.

We want the previous equation to hold for all $1 \leq i \leq N - 1$. Performing a union bound, we have that this is true with probability at least $1 - (N-1)\exp(-c_2 \log^2 N) \geq 1 - \exp(-c_3 \log^2 N)$ over $W$ and $z$. Thus, with such probability, we have

$$\begin{aligned}
\|\Phi_{-1}\tilde{\varphi}(z)\|_2 &\leq \sqrt{N-1} \max_i \left|\varphi(z_{i+1})^\top \tilde{\varphi}(z)\right| \\
&= \mathcal{O}\left(\sqrt{dk}\sqrt{N}\log^2 N\right) = o(dk),
\end{aligned} \tag{157}$$

where the last step follows from Assumption 6.1. $\qquad\square$

**Lemma E.8.** *We have*

$$\left| \frac{\tilde{\varphi}(z_1^m)^\top \tilde{\varphi}(z_1) - \tilde{\varphi}(z_1^m)^\top P_{\Phi_{-1}} \tilde{\varphi}(z_1)}{\left\| \tilde{\varphi}(z_1) - P_{\Phi_{-1}} \tilde{\varphi}(z_1) \right\|_2^2} - \frac{\tilde{\varphi}(z_1^m)^\top \tilde{\varphi}(z_1)}{\left\| \tilde{\varphi}(z_1) \right\|_2^2} \right| = o(1), \tag{158}$$

*with probability at least $1 - N \exp(-c \log^2 k) - \exp(-c \log^2 N)$ over $Z$, $x$ and $W$, where $c$ is an absolute constant. With the same probability, we also have*

$$\tilde{\varphi}(z_1^m)^\top \tilde{\varphi}(z_1) - \tilde{\varphi}(z_1^m)^\top P_{\Phi_{-1}} \tilde{\varphi}(z_1) = \Theta(dk), \qquad \left\| \tilde{\varphi}(z_1) - P_{\Phi_{-1}} \tilde{\varphi}(z_1) \right\|_2^2 = \Theta(dk). \tag{159}$$

*Proof.* Notice that, with probability at least $1 - \exp(-c \log^2 N) - \exp(-c \log^2 k)$ over $W$ and $z_1$, we have both

$$\tilde{\varphi}(z_1^m)^\top \tilde{\varphi}(z_1) = \Theta(dk) \qquad \left\| \tilde{\varphi}(z_1) \right\|_2^2 = \Theta(dk). \tag{160}$$

by the second statement of Lemma E.6. Furthermore,

$$\begin{aligned}
\left| \tilde{\varphi}(z_1^m)^\top P_{\Phi_{-1}} \tilde{\varphi}(z_1) \right| &= \left| \tilde{\varphi}(z_1^m)^\top \Phi_{-1}^\top K_{-1}^{-1} \Phi_{-1} \tilde{\varphi}(z_1) \right| \\
&\leq \left\| \Phi_{-1} \tilde{\varphi}(z_1^m) \right\|_2 \lambda_{\min}(K_{-1})^{-1} \left\| \Phi_{-1} \tilde{\varphi}(z_1) \right\|_2 \\
&= o(dk) \mathcal{O}\left( \frac{1}{dk} \right) o(dk) = o(dk),
\end{aligned} \tag{161}$$

where the third step is justified by Lemmas E.1 and E.7, and holds with probability at least $1 - N e^{-c \log^2 k} - e^{-c \log^2 N}$ over $Z$, $x$, and $W$. A similar argument can be used to show that $\left\| P_{\Phi_{-1}} \tilde{\varphi}(z_1) \right\|_2^2 = o(dk)$, which, together with (161) and (160), and a straightforward application of the triangle inequality, provides the thesis. $\quad\square$

**Lemma E.9.** *We have*

$$\left| \frac{\varphi(z_1^m)^\top P_{\Phi_{-1}}^\perp \varphi(z_1)}{\left\| P_{\Phi_{-1}}^\perp \varphi(z_1) \right\|_2^2} - \frac{\tilde{\varphi}(z_1^m)^\top P_{\Phi_{-1}}^\perp \tilde{\varphi}(z_1)}{\left\| P_{\Phi_{-1}}^\perp \tilde{\varphi}(z_1) \right\|_2^2} \right| = o(1), \tag{162}$$

*with probability at least $1 - N \exp(-c \log^2 k) - \exp(-c \log^2 N)$ over $Z$, $x$ and $W$, where $c$ is an absolute constant.*

*Proof.* If $\mu_0' = 0$, the thesis is trivial, as $\varphi \equiv \tilde{\varphi}$. If $\mu_0' \neq 0$, we can apply Lemma E.5, and the proof proceeds as follows.

First, we notice that the second term in the modulus in the statement corresponds to the first term in the statement of Lemma E.8. We will condition on the result of Lemma E.8 to hold until the end of the proof. Notice that this also implies

$$\tilde{\varphi}(z_1^m)^\top P_{\Phi_{-1}}^\perp \tilde{\varphi}(z_1) = \Theta(dk), \qquad \left\| P_{\Phi_{-1}}^\perp \tilde{\varphi}(z_1) \right\|_2^2 = \Theta(dk), \tag{163}$$

with probability at least $1 - N \exp(-c \log^2 k) - \exp(-c \log^2 N)$ over $Z$, $x$, and $W$. Due to Lemma E.5, we jointly have

$$\left\| P_{\Phi_{-1}}^\perp (z_1 \otimes \mathbf{1}_k) \right\|_2 = o(\sqrt{dk}), \qquad \left\| P_{\Phi_{-1}}^\perp (z_1^m \otimes \mathbf{1}_k) \right\|_2 = o(\sqrt{dk}), \tag{164}$$

with probability at least $1 - \exp(c \log^2 N)$ over $Z_{-1}$ and $W$. Also, by Lemma C.3 and Assumption 5.2, we jointly have

$$\left\| P_{\Phi_{-1}}^\perp \varphi(z_1^m) \right\|_2 \leq \left\| \varphi(z_1^m) \right\|_2 = \left\| z_1^m \right\|_2 \left\| \phi'(W z_1^m) \right\|_2 = \mathcal{O}\left( \sqrt{dk} \right), \tag{165}$$

and

$$\left\| P_{\Phi_{-1}}^\perp \tilde{\varphi}(z_1) \right\|_2 \leq \left\| \tilde{\varphi}(z_1) \right\|_2 = \left\| z_1 \right\|_2 \left\| \tilde{\phi}'(W z_1) \right\|_2 = \mathcal{O}\left( \sqrt{dk} \right), \tag{166}$$

with probability at least $1-\exp(-c_1\log^2 N)$ over $W$. We will condition also on such high probability events ((164), (165), (166)) until the end of the proof. Thus, we can write

$$
\begin{aligned}
\Big| \varphi(z_1^m)^\top P_{\Phi_{-1}}^\perp \varphi(z_1) &- \tilde\varphi(z_1^m)^\top P_{\Phi_{-1}}^\perp \tilde\varphi(z_1) \Big| \\
&\le \Big| \varphi(z_1^m)^\top P_{\Phi_{-1}}^\perp \left( \varphi(z_1) - \tilde\varphi(z_1) \right) \Big| + \Big| \left( \varphi(z_1^m) - \tilde\varphi(z_1^m) \right)^\top P_{\Phi_{-1}}^\perp \tilde\varphi(z_1) \Big| \\
&\le \Big\| P_{\Phi_{-1}}^\perp \varphi(z_1^m) \Big\|_2 \Big\| P_{\Phi_{-1}}^\perp \left( z_1 \otimes \mu_0 \mathbf{1}_k \right) \Big\|_2 + \Big\| P_{\Phi_{-1}}^\perp \tilde\varphi(z_1) \Big\|_2 \Big\| P_{\Phi_{-1}}^\perp \left( z_1^m \otimes \mu_0 \mathbf{1}_k \right) \Big\|_2 = o(dk),
\end{aligned}
\tag{167}
$$

where in the last step we use (164), (165), and (166). Similarly, we can show that

$$
\begin{aligned}
\left| \Big\| P_{\Phi_{-1}}^\perp \varphi(z_1) \Big\|_2 - \Big\| P_{\Phi_{-1}}^\perp \tilde\varphi(z_1) \Big\|_2 \right| &\le \Big\| P_{\Phi_{-1}}^\perp \varphi(z_1) - P_{\Phi_{-1}}^\perp \tilde\varphi(z_1) \Big\|_2 \\
&\le \Big\| P_{\Phi_{-1}}^\perp \left( z_1 \otimes \mu_0 \mathbf{1}_k \right) \Big\|_2 = o(\sqrt{dk}).
\end{aligned}
\tag{168}
$$

By combining (163), (167), and (168), the desired result readily follows. $\qquad\square$

Finally, we are ready to give the proof of Theorem 6.3.

*Proof of Theorem 6.3.* We have

$$
\begin{aligned}
\left| \frac{\varphi(z_1^m)^\top P_{\Phi_{-1}}^\perp \varphi(z_1)}{\Big\| P_{\Phi_{-1}}^\perp \varphi(z_1) \Big\|_2^2} - \alpha \frac{\sum_{l=1}^{+\infty} \mu_l'^2 \alpha^i}{\sum_{l=1}^{+\infty} \mu_l'^2} \right| &\le \left| \frac{\varphi(z_1^m)^\top P_{\Phi_{-1}}^\perp \varphi(z_1)}{\Big\| P_{\Phi_{-1}}^\perp \varphi(z_1) \Big\|_2^2} - \frac{\tilde\varphi(z_1^m)^\top P_{\Phi_{-1}}^\perp \tilde\varphi(z_1)}{\Big\| P_{\Phi_{-1}}^\perp \tilde\varphi(z_1) \Big\|_2^2} \right| \\
&\quad + \left| \frac{\tilde\varphi(z_1^m)^\top \tilde\varphi(z_1) - \tilde\varphi(z_1^m)^\top P_{\Phi_{-1}} \tilde\varphi(z_1)}{\Big\| \tilde\varphi(z_1) - P_{\Phi_{-1}} \tilde\varphi(z_1) \Big\|_2^2} - \frac{\tilde\varphi(z_1^m)^\top \tilde\varphi(z_1)}{\| \tilde\varphi(z_1) \|_2^2} \right| \\
&\quad + \left| \frac{\tilde\varphi(z_1^m)^\top \tilde\varphi(z_1)}{\| \tilde\varphi(z_1) \|_2^2} - \alpha \frac{\sum_{l=1}^{+\infty} \mu_l'^2 \alpha^i}{\sum_{l=1}^{+\infty} \mu_l'^2} \right| \\
&= o(1),
\end{aligned}
\tag{169}
$$

where the first step is justified by the triangle inequality, and the second by Lemmas E.9, E.8, and E.6, and it holds with probability at least $1 - N\exp(-c\log^2 k) - \exp(-c\log^2 N)$ over $Z$, $x$, and $W$. $\qquad\square$

# F  PROOF OF (13)

We want to characterize
$$
\mathcal{L} := \mathbb{E}_{z_1}\left[ \left( f(z_1^m, \theta^*) - g_1 \right)^2 \right],
\tag{170}
$$
where $z_1^m = [x, y_1]$ and $x \sim \mathcal{P}_X$. Assume that the labels and the predictor are balanced, namely, $\mathbb{E}_x[f(z_1^m, \theta_{-1}^*)] = 0$ and $\bar g := \mathbb{E}_{x_1}[g_1] = 0$. Then,

$$
\begin{aligned}
\mathbb{E}_x[\mathcal{L}] &= \mathbb{E}_{x,x_1,y_1}\left[ \left( f(z_1^m, \theta^*) - g_1 \right)^2 \right] \\
&= \mathbb{E}_{x,x_1,y_1}\left[ \left( f(z_1^m, \theta_{-1}^*) + \gamma_\varphi \left( g_1 - f(z_1, \theta_{-1}^*) \right) - g_1 \right)^2 \right] \\
&= \mathbb{E}_{x,x_1,y_1}\left[ \left( f(z_1^m, \theta_{-1}^*) - g_1 \right)^2 \right] + \gamma_\varphi^2 \mathbb{E}_{x_1,y_1}\left[ \left( g_1 - f(z_1, \theta_{-1}^*) \right)^2 \right] \\
&\quad + 2\gamma_\varphi \mathbb{E}_{x_1,y_1}\left[ \left( \mathbb{E}_x\left[ f(z_1^m, \theta_{-1}^*) - g_1 \right] \right) \left( g_1 - f(z_1, \theta_{-1}^*) \right) \right] \\
&= \mathbb{E}_x[\mathcal{L}_0] + \gamma_\varphi^2 \mathbb{E}_{x_1,y_1}\left[ \left( g_1 - f(z_1, \theta_{-1}^*) \right)^2 \right] - 2\gamma_\varphi \mathbb{E}_{x_1,y_1}\left[ g_1 \left( g_1 - f(z_1, \theta_{-1}^*) \right) \right] \\
&\ge \mathbb{E}_x[\mathcal{L}_0] + \gamma_\varphi^2 \mathcal{R}_{Z_{-1}} - 2\gamma_\varphi \sqrt{\mathcal{R}_{Z_{-1}}} \sqrt{\mathrm{Var}\left( g_1 \right)},
\end{aligned}
\tag{171}
$$

where $\mathcal{L}_0 = \mathbb{E}_{x_1,y_1}\left[ \left( f(z_1^m, \theta_{-1}^*) - g_1 \right)^2 \right]$.

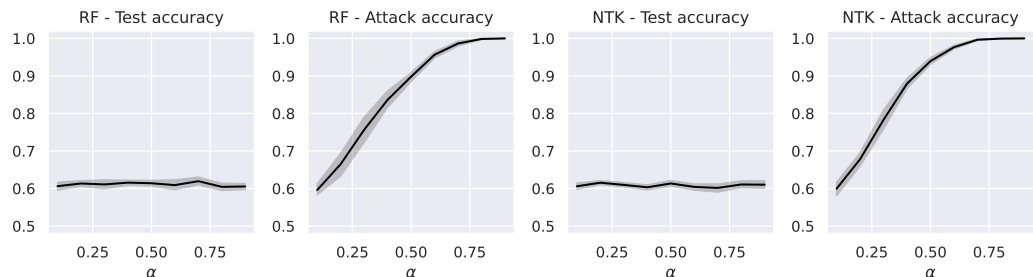

Figure 6: Test and attack accuracies as a function of $\alpha$. We consider RF (first and second plot) and NTK (third and fourth plot) models trained on a synthetic dataset. The settings are the same as in Figures 3 and 4, and we use a ReLU activation function. The number of training samples is fixed to $N = 200$.

## G  ADDITIONAL EXPERIMENTS

In Figure 6, the test and attack accuracy are plotted as a function of $0 < \alpha < 1$. While the test accuracy does not depend on $\alpha$, the accuracy of the attack monotonically grows with it. This is in agreement with the results of Theorems 5.4 and 6.3.

