# OpenReview forum: "Privacy at Interpolation: Precise Analysis for Random and NTK Features"
_ICLR.cc/2024/Conference — Submitted to ICLR 2024_

### Official Review · Reviewer_NsZa · 2023-10-31

**Soundness:** 4 excellent
**Presentation:** 3 good
**Contribution:** 2 fair
**Rating:** 6
**Confidence:** 2

**Summary:**

The paper discusses a specific form of reconstruction attack in which part of the input is masked (substituted with other data from the same distribution) and the model predicts the label. The model is a generalized linear model trained to minimize quadratic loss. With a lot of additional restrictive assumptions they derive a relationship between the model stability (sensitivity to removal of one point) and privacy (difficulty in reconstructing the true label of the masked input).

**Strengths:**

Aside from some typos and slight conceptual inclarities, the mathematical exposition is crystal clear. The setup and results are to my knowledge original (although I don't have an encyclopedic understanding of this kind of theory). The insight from the theorems is mildly interesting (but as I understand applies in a very narrow setting, and also is not really surprising, see weaknesses).

**Weaknesses:**

My main problem with the paper is that it seems to be an extremely limited scenario: linear model with squared error and invertible kernel. Why is this kind of model of use in the current era? The attack also seems quite specific. A narrow example is given (a person in a compromising environment) but do the conclusions of your method apply to more general forms of realistic attacks? How?

Second, I feel that differential privacy has been generally accepted as the gold standard for privacy, and there are carefully reasoned arguments for this choice. You use the word "privacy guarantee". Why should we be interested in this new notion of privacy? Could you relate your ideas more to DP?

The intro says "the accuracy of the attack grows with generalization error". If this is the main point of the paper, it seems, if not trivial, not especially surprising. A model that can perfectly fit the data and has high generalization error clearly can easily memorize arbitrary points, which means privacy is low and the attack should be easy.

Minor comments:
* I wouldn't say most popular applications achieve "0 training loss"; *maybe* "0 training error"
* typo "explicitely"
* Typo in citation "Milad et al." (should be "Nasr. et al")
* "$g_z$ denotes the ground-truth label of the test sample" Before you just said that $g_i$ is a function of sample $z_i$, maybe clearer to just define a label function $g(z)$.

**Questions:**

Questions already mentioned in Weaknesses section.

---

> ### Author Response · Authors · 2023-11-16
> **Response to Reviewer NsZa**
>
> We thank the reviewer for the comments on our work. We have fixed the minor issues and typos in the revised version, and we address the weaknesses below.
>
> - - - -
>
> *(1) My main problem with the paper is that it seems to be an extremely limited scenario: linear model with squared error and invertible kernel. Why is this kind of model of use in the current era? The attack also seems quite specific. A narrow example is given (a person in a compromising environment) but do the conclusions of your method apply to more general forms of realistic attacks? How?*
>
> The study of generalized linear models in the over-parameterized regime constitutes a well-established line of research in deep learning theory. In particular, the behavior of RF and NTK models in this regime ($N = o(p)$) has been the object of extensive study e.g. in the context of optimization [1, 2, 3], generalization [4, 5] and robustness [6, 7].
>
>
> While it is true that state-of-the-art deep learning models may be far from the NTK regime, the theoretical analysis of such objects still provides useful insights also in practice [8, 9]. Furthermore, even if our formal attack description requires a clear distinction between useful information $x$ and background noise $y$, our analysis provides a solid theoretical foundation to the family of attacks in which the attacker has partial knowledge about the training samples, and aims to retrieve information about the rest.
>
>
> This is particularly relevant in the context of hallucinations in language models. As mentioned at the end of Section 3, sensitive information ($x$) about an individual ($y$) could be stored in a textual dataset. The attacker could guess that $y$ was mentioned in the training dataset, and can try to recover $x$ by querying the model with the prompt
>
>
> “The address of $y$ is…”
>
>
> These experiments have been considered e.g. for question-answering tasks [10]. Specifically, in [10], the authors mask the tokens containing the information which is relevant to solve the task, and they verify that the trained model can still hallucinate the correct answer on the “masked” training samples (see Table 1 of [10]).
>
>
> While the precise formulation of more realistic problems might differ, our approach offers a solid ground to mathematically formulate different instances of this general problem in machine learning.
>
> - - - -
>
> *(2) Second, I feel that differential privacy has been generally accepted as the gold standard for privacy, and there are carefully reasoned arguments for this choice. You use the word "privacy guarantee". Why should we be interested in this new notion of privacy? Could you relate your ideas more to DP?*
>
> We agree with the reviewer that our work does not provide any formal privacy guarantee according to the formalism of Differential Privacy (DP). To better represent our contribution, we have modified what before we were addressing as “privacy guarantees” to “resistance to the attack” in the revised version of our work. In this regard, as suggested by Reviewer 4po4, we have also modified the first part of the title, replacing “Privacy at Interpolation” with “Stability and Information Retrieval”, as well as several sentences in the abstract, introduction, conclusions and right after the statements of the main theorems. We believe that these modifications better frame our technical contributions. We consider the analysis of DP training methods using the formalism developed in this work as an exciting avenue for future research.
>
> Let us conclude by addressing directly the question of the reviewer: “Why should we be interested in this new notion of privacy?” The problem of information retrieval in over-parameterized ML models is pressing in practice. Our analysis aims to characterize those situations in which DP training is not deployed, but the application would still benefit from being resistant to black-box information retrieval attacks. This is prominent in the case of LLMs [10, 11] (see also our response to the previous point) and in individual recognition in computer vision (see also Section 3 of the paper).

---

> ### Author Response · Authors · 2023-11-16
> **Response to Reviewer NsZa - Part 2**
>
> *(3) The intro says "the accuracy of the attack grows with generalization error". If this is the main point of the paper, it seems, if not trivial, not especially surprising. A model that can perfectly fit the data and has high generalization error clearly can easily memorize arbitrary points, which means privacy is low and the attack should be easy.*
>
>
> The reviewer is right in pointing out that a model that can perfectly fit the data and has high generalization error could be simply memorizing arbitrary points. This, in fact, also suggests that the attack *should* be easy, and it is in line with the empirical evidence in [10], where overfitting is seen to correlate with the success of the recovery attack. The results of our work are a **formal mathematical description of this phenomenon**, through the analysis of the concentration of the feature alignment to a strictly positive quantity, in the settings of RF and NTK regression.
>
>
> As described by Theorems 5.4 and 6.3 and shown in Figures 3-5, our analysis also links the power of the attack to the specific choice of the model and its activation function. This connection (exemplified in the second subplot of Figures 3 and 4, as well as in Figure 5 added in the revision) is highly non-intuitive, and it comes purely from our analysis.
>
>
> To conclude, as mentioned in Section 7, characterizing the concentration of the feature alignment of more complex feature maps would allow to compare different models and establish which of those are the most resistant to recovery attacks, thus allowing a principled design of privacy-preserving ML models.
>
>
> - - - -
>
>
> [1] Mahdi Soltanolkotabi, Adel Javanmard, and Jason D Lee. Theoretical insights into the optimization landscape of over-parameterized shallow neural networks. IEEE Transactions on Information Theory, 65(2):742–769, 2018.
>
> [2] Zeyuan Allen-Zhu, Yuanzhi Li, and Zhao Song. A convergence theory for deep learning via overparameterization. ICML, 2019.
>
> [3] Simon S. Du, Jason D. Lee, Haochuan Li, Liwei Wang, and Xiyu Zhai. Gradient descent finds global minima of deep neural networks. ICML, 2019.
>
> [4] Andrea Montanari and Yiqiao Zhong. The interpolation phase transition in neural networks: Memorization and generalization under lazy training. The Annals of Statistics, 2022.
>
> [5] Zhichao Wang, Yizhe Zhu. Overparameterized random feature regression with nearly orthogonal data. AISTATS, 2023
>
> [6] Zhenyu Zhu, Fanghui Liu, Grigorios Chrysos, and Volkan Cevher. Robustness in deep learning: The good (width), the bad (depth), and the ugly (initialization). NeurIPS, 2022.
>
> [7] Elvis Dohmatob. Fundamental tradeoffs between memorization and robustness in random features and neural tangent regimes. arXiv preprint arXiv:2106.02630, 2022.
>
> [8] Noel Loo, Ramin Hasani, Alexander Amini, and Daniela Rus. Evolution of neural tangent kernels under benign and adversarial training. NeurIPS, 2022.
>
> [9] Nikolaos Tsilivis and Julia Kempe. What can the neural tangent kernel tell us about adversarial robustness? NeurIPS, 2022.
>
> [10] Simone Bombari, Alessandro Achille, Zijian Wang, Yu-Xiang Wang, Yusheng Xie, Kunwar Yashraj, Singh, Srikar Appalaraju, Vijay Mahadevan, and Stefano Soatto. Towards differential relational privacy and its use in question answering. arXiv preprint arXiv:2203.16701, 2022.
>
>
> [11] Nicholas Carlini, Florian Tramer, Eric Wallace, Matthew Jagielski, Ariel Herbert-Voss, Katherine Lee, Adam Roberts, Tom B. Brown, Dawn Xiaodong Song, Ulfar Erlingsson, Alina Oprea, and Colin Raffel. Extracting training data from large language models. In USENIX Conference on Security Symposium, 2021.

---

> ### Comment · Reviewer_NsZa · 2023-11-18
> **acknolwedgement of rebuttal**
>
> Thank you for the detailed rebuttal, and for amending the paper so that it doesn't confusingly conflate stability with privacy.
>
> I still feel that the scenario is limited, and the conclusion is not too surprising. But I acknowledge that there is a significant research community interested in this sort of theory. In the spirit of not letting the review too strongly reflect my own biases, I will raise the score to 6, noting that my confidence is still only 2. If the other two, more confident, reviewers are willing to argue for acceptance, then I will not object.

---

### Official Review · Reviewer_96SU · 2023-10-31

**Soundness:** 4 excellent
**Presentation:** 3 good
**Contribution:** 3 good
**Rating:** 6
**Confidence:** 3

**Summary:**

The paper studies the safety of ERM model under interpolation. Specifically, the authors consider the settings where the adversary has access to the masked sample of the actual data used by the deep learning model. They characterize the effectiveness of this attack via two quantities: model stability with respect to the training sample and the alignment between the masked sample used by the attacker and the actual data.

**Strengths:**

- The paper provides quite an interesting insight into the privacy of deep learning models. While most previous works focus on stability under the lens of differential privacy, this work focuses more on how strong some certain attacks are depending on the resources available to the adversary. While its main conclusions are pretty much in line with previous work (a model that generalizes well is less affected by attackers), the paper still has a significant enough contribution by quantifying the level of impact of feature alignments and stability in specific settings such as Random Features and NTK.

- The paper shows that ERM model that generalizes well has some intrinsic privacy protection.

- The paper is pretty well-written overall and the proof sketch is appreciated since the detailed analysis is fairly dense and not easy to follow.

**Weaknesses:**

- The paper would probably benefit from having a section that describes and discusses the results of the experiments. Though the results in sections 5 and 6 are pretty interesting, they are also quite similar. Maybe the authors can consider cutting the proof sketch of section 6 and adding some discussion on the results of the experiment.

- I also think some discussion on the results of Theorem 5.4 and Theorem 6.3 are needed. At first glance, the main conclusion that we can get from the results is the feature alignment value would converge to some constant that depends on the norm of the background $d_y$ (which is easily interpretable) and the $l-$th Hermite coefficient of the activation function (which is not that intuitive).

**Questions:**

- In the experiments, the activation function that is composed of the Hermite polynomial is only used for experiments with synthetic data, is it possible to run that activation on real datasets? Also, how did the author create the synthetic data?

- I think the experiments right now mainly support that generalization improves privacy but it's quite hard to see where the feature alignments part comes into play here. Maybe the authors can try varying the level of alignment or $\alpha = d_y/d$ to see how it affects the attack's efficiency.

- I'm probably missing something but if the feature alignment $F_{RF}$ is in $[-1,1]$ and $0 < \gamma_{RF} <1$, the results in 5.4 doesn't seem to convey too much information? The result basically says these 2 values are a constant away from each other.

---

> ### Author Response · Authors · 2023-11-16
> **Response to Reviewer 96SU**
>
> We thank the Reviewer for the positive evaluation of our work. We address all comments below.
>
> - - - -
>
> *(Weaknesses) The paper would probably benefit from having a section that describes and discusses the results of the experiments. [...] I also think some discussion on the results of Theorem 5.4 and Theorem 6.3 are needed.*
>
> As suggested, we have included the following paragraph at the end of Section 6, providing more intuition both on our experimental and theoretical results:
>
> “As (12) suggests, if the value of $\mathcal F_\varphi(z_1, z_1^m)$ converges to 0, no information can be recovered by the attacker.
> However, our Theorems 5.4 and 6.3 show that, for RF and NTK models respectively, a strictly positive geometric overlap ($\alpha > 0$) guarantees a strictly positive feature alignment. This in turn guarantees the attacker the possibility to recover information about the original data, as long as the generalization error is not 0, as discussed in (10).
> Our experiments both on synthetic and real datasets confirm these findings. First, in Figures 3, 4 and 5, we see a decrease in the attack accuracy as generalization improves, as discussed in Section 4. This behaviour is displayed also by neural networks, as shown in Figure 1. Additionally, our experiments confirm the characterizations of $\gamma_{\rm RF}$ ($\gamma_{\rm NTK}$) in terms of $\alpha$ and the Hermite coefficients of the activation (its derivative). The expected dependence is displayed also on real datasets (see Figure 5), where functions with higher order Hermite coefficients lead to models that are more resistant to the attack.”
>
> To make space for it, as suggested by the reviewer, we have moved the sketch of the proof for NTK regression at the beginning of Appendix E.
>
> - - - -
>
> *(Question 1) In the experiments, the activation function that is composed of the Hermite polynomial is only used for experiments with synthetic data, is it possible to run that activation on real datasets? Also, how did the author create the synthetic data?*
>
> Yes, we can verify the dependence of the attack accuracy on real data. As suggested by the reviewer, we have repeated the experiments for NTK regression on MNIST and CIFAR-10 with two different activation functions, $\phi_2$ and $\phi_8$, so that their derivatives are respectively $\phi_2’ = h_0 + h_1$ and $\phi_8’ = h_0 + h_7$, where $h_i$ denotes the $i$-th Hermite polynomial.
>
> The second activation puts more weight on higher order Hermite coefficients and, according to our analysis, it should lead to models that are more resistant to the recovery attack. This is verified in Figure 5 of the revision (see Section 6), which allows to appreciate a trade-off between test accuracy and attack accuracy given by the choice of the activation function, for both datasets.
>
> Previous experiments were performed on synthetic datasets, where the samples are taken i.i.d. from a standard Gaussian distribution and the ground truth label is set to be $g_i = sign(u^\top x_i)$, for a fixed vector $u \in \mathbb R^{d_x}$. We refer to the caption of Figure 3 for these details.
>
> - - - -
>
> *(Question 2) I think the experiments right now mainly support that generalization improves privacy but it's quite hard to see where the feature alignments part comes into play here. Maybe the authors can try varying the level of alignment or $\alpha=d_y/d to see how it affects the attack's efficiency.*
>
> We would like to thank the reviewer for the question and the helpful suggestion. First of all, let us point out that the role of $\alpha$ and of the Hermite coefficients of the activation can be appreciated in the second subplot of Figures 3 and 4: $\alpha=1/2$ (red lines) leads to a more successful attack than $\alpha=1/4$ (black lines); an activation function with dominant low-order Hermite coefficients leads to a more successful attack; neither $\alpha$ nor the specific choice of the activation influence much the test accuracy. This experimentally confirms the findings of our main theoretical results.
>
> In Figure 5 of the revision (also discussed above), the effect of the activation on the accuracy of the attack is displayed for MNIST and CIFAR-10.
>
> Finally, to address the reviewer’s comment, we have performed an additional experiment with synthetic data in which we plot the test and attack accuracies as a function of $\alpha$. The results are presented in Figure 6 contained in Appendix G: the test accuracy does not depend on $\alpha$, while the accuracy of the attack monotonically grows with $\alpha$. This is again in agreement with the results of Theorem 5.4 and 6.3.

---

> ### Author Response · Authors · 2023-11-16
> **Response to Reviewer 96SU - Part 2**
>
> *(Question 3) I'm probably missing something but if the feature alignment $F_{\rm RF}$ is in $[-1, 1]$ and $0<\gamma_{\rm RF}<1$, the results in 5.4 doesn't seem to convey too much information? The result basically says these 2 values are a constant away from each other.*
>
> The role of Theorem 5.4 is to prove that $\mathcal F(z_1, z_1^m)$ concentrates to a constant bounded away from 0. In principle, the value of $\mathcal F(z_1, z_1^m)$ could converge to 0. In this case, the recovery attack wouldn’t allow the attacker to retrieve any information, as suggested by (12).
>
> In Theorem 5.4 and 6.3, we show that, for RF and NTK models respectively, a strictly positive geometric overlap ($\alpha > 0$) guarantees a strictly positive feature alignment. This in turn guarantees the attacker the possibility to recover information about the original data, as long as the generalization error is not 0, as discussed in (10).
>
> We clarify this point in the new discussion paragraph at the end of Section 6 of the revision.

---

> > ### Comment · Reviewer_96SU · 2023-11-18
> >
> > Thank you for the responses! I'll keep my score and recommend the paper for acceptance.

---

### Official Review · Reviewer_4po4 · 2023-11-01

**Soundness:** 3 good
**Presentation:** 3 good
**Contribution:** 2 fair
**Rating:** 6
**Confidence:** 3

**Summary:**

The paper analyzes the stability of generalized linear regression, and derives bounds on it, for two classes of models: random features, and two-layer networks in the NTK regime.
The paper also attempts to make a connection between these bounds and the success rates of a certain type of attacks, in which the attacker has partial access to the training example (e.g. background of an image), and seeks to recover the label.
Numerical experiments are provided, and focus on measuring the attack's accuracy and the model's generalization, showing the two are anti-correlated.

**Strengths:**

The main strength in the paper are the bounds it provides on stability of generalized linear models in the two settings of interest (random features and NTK). These bounds appear to be new. Lemma 4.1 decomposes stability into a term that involves the hat matrix (that the paper calls "feature alignment") and a term related to the model's generalization.

The paper is clearly written and the assumptions clearly stated and explained (although it would have been nice to be more upfront about certain assumptions early on, e.g. invertibility of the kernel, which requires $p \geq N$, a restrictive assumption).

The experiments give a nice illustration of the intuitive claim that attacker's success decreases with generalization.
I also commend the authors for including their code with clear instructions for reproducing the results. They clearly put effort into this.

**Weaknesses:**

The main weakness of the paper is its attempt to characterize "privacy". This is discussed in the paragraph "Generalization and privacy", and the only argument made there is that the attacker's recovery loss can be bounded below (eq. 12) by a quantity that depends on $\gamma_\phi \sqrt{R_{Z_{-1}}}$ (where $R$ is a generalization term, and $\gamma_\phi$ is the feature alignment term that the paper proceeds to bound).
The issue is that this lower bound ignores several terms, some of which are important (in particular the variance of $f(z_1^m, \theta^*)$. The lower bound of eq (12) is inadequate to really characterize the attacker's power.

Furthermore, there is *not a single formal result on privacy* stated in the paper. This was very surprising to me, given the many claims made about privacy guarantees. To cite a few:
- "Our analysis provides the first theoretical privacy guarantees for ERM algorithms interpolating the data."
- "The combination of Theorem 5.4 and Lemma 4.1 shows the proportionality between stability and privacy."
- "We provide a quantitative characterization of the accuracy of a family of powerful information recovery attacks"

I disagree with all of these claims. The paper only gives an informal, qualitative argument about attack success, together with nice experiments. There are no formal guarantees about privacy.
In summary, I think this is a good paper on stability analysis. I don't think it's a good paper on privacy.

Another area that should be improved is connecting the results with prior work. There is a brief mention of related work on stability in Section 2, but no further attempt is made to compare/contrast the results in this paper (in particular Lemma 4.1). Beside, stability in linear regression is a classic topic, it goes back much further, see for example [1]. The projector matrix defined in the paper is known as the hat matrix, and its role in stability is well studied. For example [1] gives a similar expression of stability involving the hat matrix. Though not identical to Lemma 4.1, they are closely related, and it's important to make that connection and discuss differences. I invite the authors to do a more careful review of the topic.

[1] Hoaglin and Welsch, The Hat Matrix in Regression and ANOVA. The American Statistician, 1978.

**Questions:**

- Would the authors be willing to recast the paper as a study of stability rather than privacy? I'd be more inclined to accept the paper in that case. (in particular, removing all claims about quantitative or formal privacy guarantees, removing privacy from the title, etc. It's fine to keep the empirical results.)
- Can the authors carefully review work on stability in linear regression? In particular connections to [1] and related work.

===== post rebuttal =====
I thank the reviewer for their detailed response, for exploring the connection with [1], and for their willingness to rework the presentation.
I have raised my score to 6.

---

> ### Author Response · Authors · 2023-11-16
> **Response to Reviewer 4po4**
>
> We thank the reviewer for the careful analysis of our work. We address all comments below.
>
> - - - -
>
> *(1) The lower bound of eq (12) is inadequate to characterize the attacker’s power*
>
> While it is true that the bound in (12) is not necessarily tight (because of the variance of $f(z_1^m, \theta^*)$), it still provides a worst-case estimate of the recovery power of the attacker. Specifically, if the generalization error is small, then provably the loss of the attacker cannot be better than random guessing.
>
> To address the comment of the reviewer, in the revision we also provide another bound that does not neglect the variance of $f(z_1^m, \theta^*)$. By assuming that both the labels and the predictor are balanced ($\mathbb E_{x_1}[g_1]=0$ and $\mathbb E_x[f(z_1^m, \theta^*_{-1})]=0$), we have
>
> $$
> \mathbb E_x[\mathcal L]\ge \mathbb E_x[\mathcal L_0] +\gamma_\varphi^2\mathcal R_{Z_{-1}}-2\gamma_\varphi\sqrt{\mathcal R_{Z_{-1}}}\sqrt{{\rm Var}(g_1)},
> $$
>
> where $\mathcal L_0=\mathbb E_{x_1, y_1}[(f(z_1^m, \theta^*_{-1})-g_1)^2]$ denotes the loss of an attacker that has no information available on the true label $g_1$. We note that, when the generalization error is small, the RHS is close to $\mathbb E_x[\mathcal L_0] -2\gamma_\varphi\sqrt{\mathcal R_{Z_{-1}}}\sqrt{{\rm Var}(g_1)}$, which is similar to the lower bound in eq (12). We have added this new bound at the end of Section 4 and deferred its proof to Appendix F.
>
> - - - -
>
> *(2) Would the authors be willing to recast the paper as a study of stability rather than privacy?*
>
> Yes, we agree with the reviewer and have revised accordingly. In particular:
> - We have removed “Privacy” from the title, replacing it with “Stability” and “Information Retrieval”.
> - We have removed the sentence “Our analysis provides the first theoretical privacy guarantees for ERM algorithms interpolating the data".
> - We have rephrased the sentence “The combination of Theorem 5.4 and Lemma 4.1 shows the proportionality between stability and privacy” as “The combination of Theorem 5.4 and (10) shows that the more the algorithm is stable (and, therefore, capable of generalizing), the smaller is the statistical dependency between the query of the attacker and the true label”.
> - We have rephrased the sentence “We provide a quantitative characterization of the accuracy of a family of powerful information recovery attacks” as “We study the accuracy of a family of powerful information recovery attacks for models trained via empirical risk minimization, in the interpolation regime”.
> - Several other changes in this direction have been made to the abstract, introduction, discussion after the main theorem statements, and conclusions.
>
>
> All these changes are in red color to facilitate the reviewing process.
>
> We would like to highlight that these modifications only concern how to interpret our technical results. The technical content of the paper remains unchanged. The only exception is the additional bound on the loss of the attacker discussed above.
>
> - - - -
>
> *(3) Can the authors carefully review work on stability in linear regression? In particular connections to [1] and related work.*
>
>
> We thank the reviewer for pointing the related work [1], which describes the role of the Hat matrix to estimate the influence of a single data point in linear regression.
>
> We remark that the Hat matrix $H := X (X^\top X)^{-1} X^\top$ can be defined when $X^\top X$ is invertible, which requires (in our notation) $p \leq N$. As we consider the overparameterized regime ($p > N$), this condition cannot be satisfied.
>
> In the underparameterized regime, following 5.5 in [1], we get (using our notation)
>
> $$f(z_1, \theta^*) - f(z_1, \theta^*_{-1}) = h_1 r_1 / (1 - h_1),$$
>
> where $h_1$ is the first element in the diagonal of $H$, and $r_1$ is the residual on the first sample, i.e.,
>
> $$r_1 = g_1 - f(z_1, \theta^*).$$
>
> At interpolation, $r_1 = 0$ and the leave-one-out stability has to be derived alternatively.
>
> We agree with the reviewer that our analysis leverages projectors having a similar form, such as $P_{\Phi} = \Phi^\top (\Phi \Phi^\top)^{-1} \Phi$. However, Lemma 4.1, which uses $P_{\Phi_{-1}}^\perp$ to describe the leave-one-out stability of overparameterized generalized linear models, is – to the best of our knowledge – novel.
>
> In the revision, we mention the related work [1] in Section 2, and then we discuss it more in detail right after the statement of Lemma 4.1.

---

### Author Response · Authors · 2023-11-16
**Main changes in the revision**

Following the suggestion of Reviewer 4po4, we have revised the paper moving the emphasis from **privacy** to **stability** and **information retrieval**, as we agree that this better frames the main contribution of our work. In particular, we have modified the first line of the title from “Privacy at Interpolation” to “Stability and Information Retrieval”. We have modified (or completely removed) all the sentences mentioned by Reviewer 4po4, and we have made a number of other edits going in the same direction to the body of the paper (specifically, in the abstract, introduction, conclusions and right after the statements of the main theorems). All such edits can be found in red color to facilitate the reviewing process.

We would like to highlight that these modifications only concern how to interpret our technical results. The technical content of the paper remains unchanged. The only exception is an additional bound on the loss of the attacker which does not neglect the variance of $f(z_1^m, \theta^*)$, see point (1) of the response to Reviewer 4po4 for details.

As suggested by Reviewer 96SU, we also provide additional experiments in the revised version. These new experiments concern:

- The role of the activation function for NTK regression with real datasets (MNIST and CIFAR-10). As suggested by Theorem 6.3, activations with more weight on higher order Hermite coefficients lead to models that are more resistant to the recovery attack. The plots are presented in Figure 5 contained in Section 6 of the revision.

- The dependence of the test and attack accuracy on the geometric overlap $\alpha$. We consider a synthetic dataset with $N = 200$ training samples, and $\alpha$ varying between 0.1 and 0.9. As expected, we see an increase in the attack accuracy with alpha, while the test accuracy remains unchanged. The results are in Figure 6 contained in Appendix G of the revision.

We hope that our response addresses all points raised by the reviewers and we remain at disposal for additional clarifications.

---

### Meta-Review · Area_Chair_bTo4 · 2023-12-12

**Metareview:**

The paper discusses a connection between generalization, leave-one-out-stability and a specific form of reconstruction attack in which part of the input is masked (substituted with other data from the same distribution) and the model predicts the label.
The model is an overparamerized linear model with random features trained to minimize quadratic loss. The theoretical analysis which is done under a number of relatively restrictive assumptions, shows a relationship between the model LOO stability (difficulty in reconstructing the true label of the masked input). This is supplemented with a experiments on small datasets. The contribution is technically novel and clearly presented. At the same time the setting is very restrictive and the overall conceptual message is mostly in line with known connections between stability and generalization.

**Justification For Why Not Higher Score:**

The conceptual message is relatively weak. But I don't mind it getting accepted

**Justification For Why Not Lower Score:**

N/A

---

### Decision · Program_Chairs · 2024-01-16

Reject